# Divergent actions of physiological and pathological amyloid-β on synapses in live human brain slice cultures

Robert I. McGeachan [1,2,3,8], Soraya Meftah [1,2,8], Lewis W. Taylor[1,2], James H. Catterson [1,2], Danilo Negro [1,2], Calum Bonthron [1,2], Kristján Holt[1,2], Jane Tulloch [1,2], Jamie L. Rose [1,2], Francesco Gobbo [1,2], Ya Yin Chang [2], Jamie Elliott [1,2], Lauren McLay [1,2], Declan King [1,2], Imran Liaquat[4], Tara L. Spires-Jones [1,2], Sam A. Booker[1,5], Paul M. Brennan [4,6,7] & Claire S. Durrant [1,2] ✉

In Alzheimer's disease, amyloid beta (Aβ) and tau pathology are thought to drive synapse loss. However, there is limited information on how endogenous levels of tau, Aβ and other biomarkers relate to patient characteristics, or how manipulating physiological levels of Aβ impacts synapses in living adult human brain. Using live human brain slice cultures, we report that $Aβ_{1-40}$ and tau release levels vary with donor age and brain region, respectively. Release of other biomarkers such as KLK-6, NCAM-1, and Neurogranin vary between brain region, while TDP-43 and NCAM-1 release is impacted by sex. Pharmacological manipulation of Aβ in either direction results in a loss of synaptophysin puncta, with increased physiological Aβ triggering potentially compensatory synaptic transcript changes. In contrast, treatment with Aβ-containing Alzheimer's disease brain extract results in post-synaptic Aβ uptake and pre-synaptic puncta loss without affecting synaptic transcripts. These data reveal distinct effects of physiological and pathological Aβ on synapses in human brain tissue.

Alzheimer's disease (AD) is a common neurodegenerative disorder that causes progressive, life-limiting dementia. Neuropathologically, it is characterised by inflammation, atrophy (including loss of synapses and neurons), and accumulation of amyloid-β (Aβ) plaques and neurofibrillary tau tangles[1,2]. Loss of synapses is the best correlate of cognitive decline, and so understanding how early changes to Aβ and tau impact synapse health will be crucial for the development of effective therapeutics[1]. Despite evidence that pathological Aβ and tau can induce synapse dysfunction in model systems[1,3–5] and reports that

novel Aβ-targeting antibodies can, modestly, slow cognitive decline[6–8], AD research has had limited translational success[1,9,10]. This lack of progress is likely due to many AD models failing to recapitulate key aspects of human brain function, including lifespan, brain size, neuronal diversity and ability to assess disease-relevant biomarkers[11–14].

Whilst significant progress has been made in characterising Aβ and tau progression in patients with AD[15], direct real-time analysis of endogenous protein levels, within the healthy human brain presents many challenges. Lumbar cerebrospinal fluid (CSF) displays lowered

[1]Centre for Discovery Brain Sciences, The University of Edinburgh, Edinburgh, UK. [2]UK Dementia Research Institute, The University of Edinburgh, Edinburgh, UK. [3]The Hospital for Small Animals, Royal (Dick) School of Veterinary Studies, The University of Edinburgh, Edinburgh, UK. [4]Department of Clinical Neuroscience, Royal Infirmary of Edinburgh, 51 Little France Crescent, Edinburgh, UK. [5]Simons Initiative for the Developing Brain, The University of Edinburgh, Edinburgh, UK. [6]Translational Neurosurgery, The Centre for Clinical Brain Sciences, The University of Edinburgh, Edinburgh, UK. [7]Cancer Research UK Brain Tumour Centre of Excellence, CRUK Edinburgh Centre, The University of Edinburgh, Edinburgh, UK. [8]These authors contributed equally: Robert I. McGeachan, Soraya Meftah. ✉e-mail: claire.durrant@ed.ac.uk

$A\beta_{1-42}$ and increased phosphorylated tau during normal ageing, which is more pronounced in *APOE* ε4 carriers or in people living with AD[16–19]. The levels of other proteins in the CSF have also shown promise as potential biomarkers for neurodegenerative disease progression, with the levels of neurogranin (a post-synaptic marker) and kallikrein-related peptidase-6 (KLK-6) levels correlating with AD severity and advanced age respectively[20–23]. However, CSF measures are an aggregate output from the brain influenced by many factors including: breakdown in the blood-brain barrier[24–26], circadian rhythms[27–29], and rates of protein production, degradation or clearance[30–32]. As such, CSF cannot provide direct information on protein levels within brain tissue or the proportion of released proteins arising from different brain regions. By contrast, PET scans, whilst unable to detect soluble forms of protein[5,33–35], permit spatiotemporal visualisation of fibrillar $A\beta$ and tau burden within the brain. PET scans in ageing adults find that whilst $A\beta$ deposition increases throughout the brain with age, and is worse in *APOE* ε4 carriers[36,37], tau specifically accumulates within the medial temporal lobe[38,39]. Post-mortem analysis of brain tissue reveals a decline in soluble $A\beta$ levels correlated with increased insoluble $A\beta$ with typical ageing or in AD[40–42]. Such findings, however, represent only the end-stage disease and not the dynamics of its' progression. Examination of brain interstitial fluid (ISF) provides a much closer representation of protein dynamics in the brain, but these studies are rarely performed due to the invasive nature of ISF sampling restricting analysis to patients with shunts or severe head injury[43,44]. Such limitations mean we still know very little about real-time $A\beta$, tau and other putative biomarker dynamics in the living human brain across the lifespan.

Understanding how living synapses respond to real-time changes in $A\beta$ also remains a significant challenge in human research. Studies in rodents[45–48], non-human primates[49] and neurons derived from human induced pluripotent stem cells (iPSCs)[50–54] show that experimentally applied oligomeric $A\beta$ binds to post-synapses and can trigger synaptotoxicity. Whilst analysis of end-stage post-mortem AD brain reveals that oligomeric $A\beta$ accumulates at the synapse[55–57], and synthetic $A\beta$ can induce loss of synaptophysin mRNA in free-floating human brain slices[58,59], direct evidence that the $A\beta$ present in AD brains is synaptotoxic to human brain tissue is lacking. To our knowledge, the effect of $A\beta$ oligomer application or pharmacologically raising endogenous $A\beta$ concentration has, due to presumed toxicity, never been studied in human patients. The effects of reducing $A\beta$ in humans has been restricted to indirect CSF, imaging, or post-mortem studies following clinical trials of $A\beta$ lowering therapies in AD[7,8,60–65]. Interestingly, lowering physiological levels of $A\beta$ in wildtype rodents can disrupt synaptic function, indicating that $A\beta$ may play both physiological and pathological roles at the synapse[3,66]. Understanding how synapses in live human brain respond to both pathological and physiological alterations to $A\beta$ will fill key knowledge gaps that could refine therapeutic design for AD.

In recent years, live human brain slice cultures (HBSCs) have been established as an exciting translational tool to investigate: physiological properties of human neurons[12,67–76], tumour environments[77–79], neurodegenerative disease[58,59,80–88], epileptic activity[89,90] and developmental disorders[91]. Free floating human brain slices have also been used to show that application of high nanomolar concentrations of synthetic $A\beta$ results in $A\beta$ being taken up by brain tissue[58] and alters mRNA expression of a number of genes including downregulation of synaptophysin[59]. As a tool, the ease of access to the culture medium provides a unique opportunity to observe how proteins released from the living human brain tissue relate to patient characteristics and disease pathogenesis.

In this work, we harness HBSCs to explore how endogenous $A\beta$ and tau release and other neurodegenerative-associated biomarkers from brain tissue are impacted by patient characteristics. We find that the levels of $A\beta_{1-40}$ release are impacted by patient age; the levels of tau, KLK-6, NCAM-1 and Neurogranin release vary by brain region; TDP-43 and NCAM-1 release is impacted by patient sex. By pharmacologically manipulating physiological levels of $A\beta$, we find that altered $A\beta$ concentration in either direction results in a loss of pre-synaptic puncta. However, there was evidence for a potentially compensatory upregulation in synaptic transcripts when physiological $A\beta$ levels are increased. By contrast, application of pathological AD-derived $A\beta$ results in $A\beta$ binding to post-synaptic compartments and a loss of pre-synaptic puncta in the absence of alterations to synaptic transcripts. Finally, we find evidence of $A\beta$ plaques and tau tangles in a subset of our samples, with levels of pathology in the slice correlating with KLK-6 expression and $A\beta_{1-42}/A\beta_{1-40}$ ratio from the culture medium. Together, our data reveals differing effects of physiological and pathological $A\beta$ on synapses and underscores the potential of HBSCs to investigate unexplored aspects of human pre-clinical AD pathology.

## Results

### Preservation of cellular diversity and function in human brain slice culturess at 7 days in vitro

Over the course of this study, we generated slice cultures from surplus neocortical tissue removed to obtain access to the tumour for debulking surgery of glioblastoma (21/42), glioma (14/42) or brain metastases (7/42), from 42 patients (Fig. 1a; Table 1). Our total cohort comprised 17 females and 25 males with an average age of 58 years old (ages ranging from 28 to 77 years old ±14 years (standard deviation)) (Table 1). The majority of samples were from the frontal (20/42) and temporal (12/42) lobes, with a subset of samples derived from parietal or occipital lobes or borderline between two lobes (10/42). For each case we screened this tissue macroscopically, looking for evidence of tumour infiltration or poor tissue quality (differentiation of white vs grey, colour of tissue, clear lamination including visible layer IV; Supplementary Fig. 1). From a subset of our cases ($n = 6$), we then established whether key cell types in the brain were preserved in HBSCs for at least 7 days in vitro (*div*). MAP2 and NeuN-positive neurons were preserved in HBSCs across the cortical layers (Fig. 1b–d). We also detected microglia in different states, using the classical pan-microglia marker Iba1[92] (Fig.1e) alongside the pyrogenic receptor P2RY12 (known to be expressed on the ramified processes of microglia[93]) (Fig. 1f). GFAP-positive astrocytes were present throughout the slice tissue and could be observed in contact with putative blood vessels (Fig. 1g). At 7 *div* we were also able to detect spontaneous synaptic activity and evoke action potentials using whole-cell patch clamp recordings of individual neurons in a subset of cases tested ($n = 5$ cases; Fig. 1h, i). We assessed how HBSCs changed from acute to 7 *div* (Supplementary Fig. 1) and found a decrease in synaptic and neuronal proteins (Synaptophysin, PSD95, PGP9.5, NR1) during the culture period (as would be expected from a resected tissue model) with no changes to Tuj1 or GAD2 (Supplementary Fig. 1). Lactate dehydrogenase release, a marker of cytotoxicity, was below 5% cytotoxicity at 7 *div* ($n = 3$; Supplementary Fig. 1).

### The impact of patient characteristics on basal release of neurodegenerative fluid biomarkers

Next, we took advantage of the biological diversity of our HBSCs, to assess whether differences in brain region, donor age, sex, or *APOE* genotype impact the basal release of key AD-associated proteins and other neurodegenerative biomarkers. Tissue from 20 different cases, ages ranging from 28 to 77, were cultured for 7 *div*, following which the culture medium was collected for analysis of $A\beta_{1-40}$, $A\beta_{1-42}$, and total tau by ELISA. KLK-6, neural cell adhesion molecule-1 (NCAM-1), neurogranin and TAR DNA-binding protein 43 (TDP-43) were quantified using a Luminex® Multiplex Assay. Endogenous release of these proteins, normalised to total protein, was readily detected in culture medium. We found a trend for a decline in $A\beta_{1-40}$ detected in the culture medium with increasing patient age (Fig. 2a), whilst the observed levels of $A\beta_{1-42}$ remained stable with age (Fig. 2b). The levels

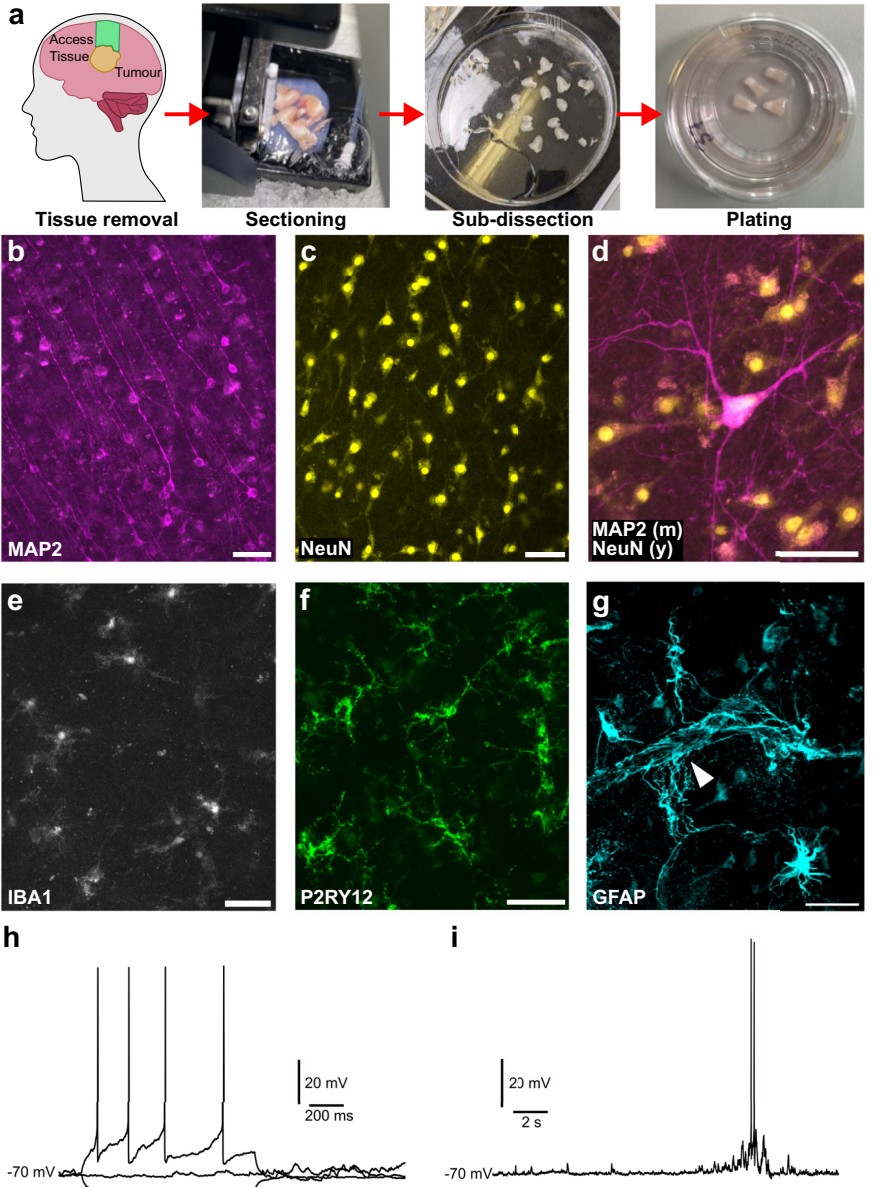

**Fig. 1 | Characterisation of human brain slice cultures. a** An overview of human brain slice culture generation, starting from access tissue removal during brain tumour surgery through to slice culture plating in dishes. **b–g** 7 *div* immuno-fluorescence images of MAP2 (**b**) NeuN (**c**), MAP2 (magenta) and NeuN (yellow) merged (**d**) Iba1 (**e**) P2RY12 (**f**) and GFAP (**g**). A white arrowhead highlights astrocyte end-feet wrapping around a putative blood vessel (**g**). Scale bars are 50 μm in length shown at the bottom right corner of the image. $N = 6$ cases were screened for IHC markers. **h**, **i** Example electrophysiological properties of slice cultures at *div* 7. **h** Responses to hyperpolarising and depolarising current injections, eliciting action potentials at depolarising currents. **i** Spontaneous synaptic properties at rest show small and large coordinated levels of synaptic drive. $N = 5$ cases were used for electrophysiological recordings.

of $A\beta_{1-40}$ were around 16 times higher than the levels of $A\beta_{1-42}$ at baseline, with an average concentration of $16.6 \pm 5.8$ and $1.1 \pm 1.9$ pM, respectively. $A\beta_{1-42}/A\beta_{1-40}$ ratio remained relatively stable with donor age (Fig. 2c). We found no correlation between patient age and the concentration of total tau protein in the medium, with an average detected concentration of $1.4 \pm 0.9$ nM (Fig. 2d). We also found no correlation with patient age and levels of KLK-6, NCAM-1, Neurogranin or TDP-43 (Fig. 3a–d). A potential explanation for changes in $A\beta_{1-40}$ release could be differences in the expression of cellular or synaptic markers. When looking at 7 *div*, no significant correlations were seen between levels of Synaptophysin, PSD95, Tuj1 or PGP9.5 in the slice tissue and $A\beta_{1-40}$, $A\beta_{1-42}$, tau or other neurodegenerative biomarker levels in culture medium (Supplementary Figs. 2 and 3). This indicates that the levels of neurons or synapses in the culture are not responsible

for the drop in $A\beta$ with age. When examining protein levels in acute tissue samples, we found a significant decrease in neuronal marker Tuj1 associated with age (Supplementary Fig. 2).

The majority of our samples for this experiment were taken from either temporal lobe (9/21) or frontal lobe (9/21), with only 3 samples taken from parietal or occipital regions (classed as "other" in this study). We found no effect of brain region on the levels of $A\beta_{1-40}$ (Fig. 2e) or $A\beta_{1-42}$ (Fig. 2f) detected in the culture medium, and no impact on the $A\beta_{1-42}/A\beta_{1-40}$ ratio (Fig. 2g). We found a significant effect of brain region for tau, with temporal lobe samples having higher levels of tau in the culture medium, when compared to samples originating from the frontal lobe (Fig. 2h). This is especially interesting in light of the fact that tau pathology in AD generally originates in the medial temporal lobe[94–96]. We also found significant effects of brain

**Table 1 | Case details for human brain slice cultures**

| Case ID | Sex | Age (rounded to nearest 5 years)*Age to nearest 1 year used in data analysis | APOE genotype | Brain region | Amyloid pathology | Tau pathology | Study used |
|---|---|---|---|---|---|---|---|
| Case 1 | Male | 70 | 3/3 | Parietal Occipital | No | No | Fig. 6 |
| Case 2 | Female | 35 | 2/4 | Frontal | No | No | Figs. 2, 3, 6, 7, Supp. Figs. 2, 3 |
| Case 3 | Female | 30 | 3/3 | Temporal | No | No | Figs. 2, 3, 6, 7 |
| Case 4 | Female | 30 | 3/3 | Frontal | No | No | Figs. 2, 3, 6, 7 |
| Case 5 | Male | 65 | 3/3 | Temporal | Plaque | Thread | Figs. 1, 2, 3, 6, 7, Supp. Figs. 2, 3 |
| Case 6 | Male | 45 | 3/3 | Frontal Parietal | No | Thread | Fig. 6, Supp. Fig. 2 |
| Case 7 | Female | 75 | 3/3 | Temporal | No | Tangle | Figs. 2, 3, 6, 7, Supp. Fig. 2 |
| Case 8 | Male | 70 | 3/3 | Frontal | N/A | N/A | Supp. Fig. 2 |
| Case 9 | Female | 75 | 3/4 | Temporal | Plaque | Tangle | Figs. 2, 3, 6, 7, Supp. Figs. 1, 2 |
| Case 10 | Male | 55 | 3/3 | Frontal | No | Thread | Figs. 2, 3, 5–7, Supp. Figs. 2, 7, 8 |
| Case 11 | Female | 50 | 3/3 | Frontal | No | No | Figs. 2–7, Supp. Figs. 1–4, 7, 8 |
| Case 12 | Male | 55 | 2/3 | Temporal | No | No | Figs. 2–7, Supp. Figs. 2, 4, 7, 8 |
| Case 13 | Female | 65 | 3/3 | Frontal | No | No | Figs. 2–7, Supp. Figs. 1–5, 7, 8 |
| Case 14 | Male | 55 | 3/3 | Parietal | No | Tangle | Figs. 2–7, Supp. Figs. 1–5, 7, 8 |
| Case 15 | Male | 75 | 3/3 | Temporal | No | Tangle | Figs. 1, 6, Supp. Fig. 2 |
| Case 16 | Male | 40 | 3/3 | Temporal | No | No | Figs. 2–4, 6, 7, Supp. Figs. 1–5 |
| Case 17 | Male | 75 | 3/4 | Occipital | Plaque | Tangle | Figs. 2–4, 6, 7, Supp. Figs. 1–5 |
| Case 18 | Male | 45 | 3/3 | Parietal | No | Thread | Figs. 1–4, 6, 7, Supp. Figs. 1–5 |
| Case 19 | Female | 40 | 3/4 | Frontal | No | No | Figs. 2–6, 7, Supp. Figs. 1–5 |
| Case 20 | Male | 75 | 3/4 | Temporal | N/A | N/A | Figs. 2–4, Supp. Figs. 1–5 |
| Case 21 | Male | 65 | 3/4 | Frontal | No | Tangle | Figs. 1–4, 6, 7, Supp. Figs. 1–5 |
| Case 22 | Male | 55 | 2/3 | Frontal | No | No | Figs. 1–4, 6, 7, Supp. Figs. 1–5 |
| Case 23 | Female | 30 | 3/3 | Temporal | No | No | Figs. 1–4, 6, 7, Supp. Figs. 1–5 |
| Case 24 | Male | 70 | 2/3 | Frontal | Plaque | No | Figs. 2–4, 6, 7, Supp. Figs. 2, 4 |
| Case 25 | Female | 55 | N/A | Parietal | N/A | N/A | Supp. Fig. 2 |
| Case 26 | Male | 65 | 2/3 | Temporal | No | Tangle | Figs. 1, 4, 6, Supp. Figs. 2, 4 |
| Case 27 | Male | 70 | 2/3 | Parietal | No | No | Figs. 1, 4, 6, Supp. Figs. 2, 4 |
| Case 28 | Female | 65 | 3/4 | Frontal | N/A | N/A | Supp. Fig. 2 |
| Case 29 | Female | 65 | 3/3 | Parietal | N/A | N/A | Supp. Fig. 2 |
| Case 30 | Male | 60 | 3/3 | Parietal | No | No | Figs. 1, 4, 6, Supp. Figs. 2, 4 |
| Case 31 | Female | 60 | 3/3 | Parietal | N/A | N/A | Fig. 4, Supp. Fig. 2 |
| Case 32 | Male | 40 | 3/3 | Frontal | No | No | Figs. 4, 6, Supp. Figs. 2, 4 |
| Case 33 | Male | 75 | 3/3 | Frontal | No | No | Figs. 4, 6, Supp. Figs. 2, 4 |
| Case 34 | Male | 35 | 3/3 | Frontal | No | No | Figs. 1, 5, 6, Supp. Figs. 2, 7 |
| Case 35 | Male | 70 | 3/3 | Frontal | No | No | Fig. 6, Supp. Fig. 2 |
| Case 36 | Female | 30 | 3/3 | Frontal | No | No | Figs. 5, 6, Supp. Figs. 2, 7 |
| Case 37 | Female | 70 | 3/3 | Frontal | Plaque | No | Figs. 4–6, Supp. Figs. 1, 2, 4, 7 |
| Case 38 | Male | 55 | 3/3 | Frontal | No | No | Figs. 4–6, Supp. Figs. 2, 4, 7 |
| Case 39 | Male | 65 | 3/3 | Temporal | No | No | Figs. 1,4–6, Supp. Figs. 1, 2, 4, 7 |
| Case 40 | Male | 50 | 3/3 | Frontal | Plaque | No | Figs. 4, 6, Supp. Figs. 1, 2, 4 |
| Case 41 | Female | 45 | 3/3 | Frontal | No | No | Figs. 4, 6, Supp. Fig. 4 |
| Case 42 | Male | 75 | 2/3 | Temporal | N/A | N/A | Supp. Fig. 2 |

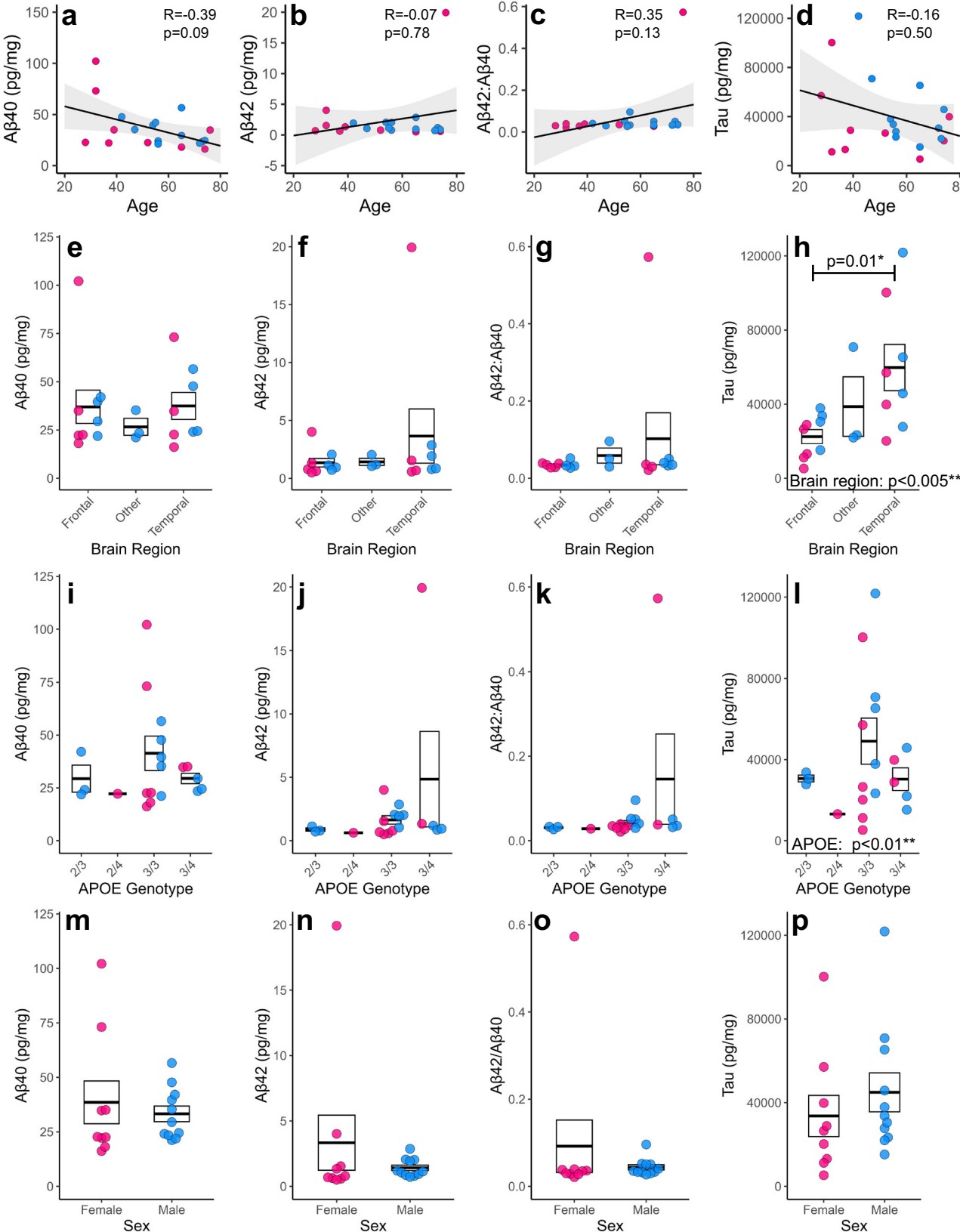

region on KLK-1, NCAM-1 and Neurogranin concentration (Fig. 3e–g), but no relationship with TDP-43 (Fig. 3h).

*APOE* polymorphic alleles are the main genetic determinant of late-onset AD risk, with individuals carrying the ε4 allele at greater risk of developing the disease, and those with ε2 having lower risk[97,98].

*APOE* genotype distribution for our samples was representative of what we would expect to see in a white British population, with ε3/ε3 being the most common genotype[98]. ε2/ε2 or ε4/ε4 genotypes were not represented in our donor samples. We found no overt impact of *APOE* genotype on the levels of Aβ$_{1-40}$, Aβ$_{1-42}$ or Aβ$_{1-42}$/Aβ$_{1-40}$ ratio

**Fig. 2 | Relationship between Aβ, tau, and patient characteristics.** Proteins of interest in the culture medium were measured by ELISA and normalised to total protein content in the medium. **a–d** Scatter plots showing the correlations between age of patient and $Aβ_{1-40}$ ($r_{s(20)} = -0.39$, $p = 0.09$) (**a**) $Aβ_{1-42}$ ($r_{s(20)} = -0.07$, $p = 0.78$) (**b**) $Aβ_{1-42}/Aβ_{1-40}$ ratio ($r_{s(20)} = 0.35$, $p = 0.13$) (**c**), and total tau ($r_{s(20)} = -0.16$, $p = 0.50$) (**d**) detected in the culture medium. Regression lines are plotted on, with shaded area representing 95% confidence intervals. Statistics: Spearman's rank correlation due to non-normal distribution, $n = 20$ human cases for above analysis. **e–h** Box and dot plots assessing the relationship between brain region and $Aβ_{1-40}$ (log, $χ^2_{(2,20)} = 1.49$, $p = 0.47$) (**e**) $Aβ_{1-42}$ (log, $χ^2_{(2,20)} = 0.86$, $p = 0.64$) (**f**) $Aβ_{1-42}/Aβ_{1-40}$ ratio (gaussian, $χ^2_{(2,20)} = 1.26$, $p = 0.53$) (**g**) and total tau (log, $χ^2_{(2,20)} = 17.04$, $p = 0.00020***$) (**h**) in the culture medium. Statistics: Generalised linear mixed models (GLMM) with format variable - Brain region + Sex + (1|Age), $n = 20$ human cases for above analysis. **i–l** Box and dot plots assessing the relationship between *APOE* genotype and $Aβ_{1-40}$ (Tukey, $χ^2_{(2,20)} = 0.34$, $p = 0.95$) (**i**) $Aβ_{1-42}$ (Tukey, $χ^2_{(2,20)} = 3.5$, p = 0.32) (**j**), $Aβ_{1-42:1-40}$ ratio (Tukey, $χ^2_{(2,20)} = 3.41$, $p = 0.33$) (**k**) and tau (log, $χ^2_{(2,20)} = 11.44$, $p = 0.01**$) (**l**) detected in the culture medium. Statistics: GLMM with format variable - APOE + Sex + (1|Age), $n = 20$ human cases for above analysis. **m–p** Box and dot plots assessing the relationship between sex and $Aβ_{1-40}$ (gaussian, $χ^2_{(1,20)} = 0.32$, $p = 0.57$) (**m**) $Aβ_{1-42}$ (Tukey, $χ^2_{(1,20)} = 0.77$, $p = 0.38$) (**n**) $Aβ_{1-42:1-40}$ ratio (log, $χ^2_{(1,20)} = 1.14$, $p = 0.29$) (**o**) and tau (log, $χ^2_{(1,20)} = 2.15$, $p = 0.14$) (**p**) detected in the culture medium. Statistics: GLMM with format variable - Sex + (1|Age), $n = 20$ human cases for above analysis. **a–p** Pink dots are for female cases and blue dots for male cases. **e–p** Box represents standard error of the mean (SEM), with thick lines representing the mean. Source data are provided as a Source Data file.

detected in the culture medium in our samples (Fig. 2i–k). There was a significant effect of *APOE* on the detection of tau in the medium (Fig. 2l), however we are underpowered to fully elucidate differences between genotypes. There was also an effect of KLK-6 release with *APOE* genotype (Fig. 3i).

Sex is a significant risk factor for AD, with almost 2/3rds of patients being women, with suggested sex-specific mechanisms playing a role beyond differences in average age of these populations[99]. In our samples, sex mostly had no impact on the concentration of specific proteins in the medium (Figs. 2 and 3), with the exception that NCAM-1 and TDP-43 levels were significantly higher in males compared to females (Fig. 3n,p).

## Bi-directional manipulation of Aβ results in loss of pre-synaptic puncta, but with divergent impacts on synaptic transcripts

Having determined that the endogenous release of AD-relevant proteins could be readily detected in our sample set, we next sought to explore whether we could pharmacologically alter Aβ in HBSCs. Previous studies in mice or primary cultures have shown that inhibition of the beta-secretase beta-site amyloid precursor protein (APP) cleaving enzyme (BACE1) significantly reduces Aβ production[64], whilst inhibition of metalloproteases reduces Aβ breakdown[100–104]. Here, we employed a repeated measures design such that tissue from the same patient was split into three independent HBSC dishes. HBSCs from the same patient were then treated with either 5 μM of the BACE1 inhibitor LY2886721, 100 μM of the metalloprotease inhibitor Phosphoramidon, or medium only control. Culture medium was collected after 7 *div*, with the measured levels of the protein of interest normalised to the corresponding medium only control. Secreted, soluble levels of $Aβ_{1-40}$ (Fig. 4a) and $Aβ_{1-42}$ (Fig. 4b) detected in the culture medium were significantly increased by Phosphoramidon treatment and reduced by BACE1 inhibition. The $Aβ_{1-42}/Aβ_{1-40}$ ratio remained unchanged regardless of treatment (Fig. 4c), suggesting the two isoforms were proportionally impacted. The concentration of tau detected in the culture medium was unchanged by treatment (Fig. 4d). Absolute data (not normalised to medium-only control) is available in (Supplementary Fig. 4). The levels of APP protein, as measured by Western blot, remained stable with treatment (Supplementary Fig. 5), confirming that BACE1 or metalloprotease inhibition acts downstream of APP to alter Aβ levels. Guanidine extraction of insoluble Aβ from the slice tissue showed no impact of Phosphoramidon or BACE1 treatment on the detected levels of $Aβ_{1-40}$ or $Aβ_{1-42}$ (Fig. 4e). There was, however, a significant treatment effect on the percentage area of Aβ within the slice when measured by MOAB-2 (pan-Aβ antibody that does not detect APP) immunostaining (Fig. 4f). Interestingly, Phosphoramidon treatment resulted in increased expression of the synaptic transcripts synaptophysin (*SYP*) and synaptotagmin 1 (*SYT1*), with *SNAP25* also having a significant effect of treatment (Fig. 4g). There was also a significant treatment effect on *IBA1* mRNA expression. The levels of *GAD2*, *GFAP* and *C1QA* mRNA remained unchanged by treatment

(Fig. 4g). Conversely, array tomography revealed a reduction in synaptophysin puncta density in response to treatment overall, suggesting manipulation of Aβ levels can alter pre-synaptic stability, with no alterations seen in PSD95 density (Fig. 4h and i). Western blots revealed that neither BACE1 inhibitor nor Phosphoramidon treatment impacted levels of the neuronal proteins Tuj1 or PGP9.5, synaptic proteins synaptophysin and PSD95, or astrocytic proteins GFAP or YKL-40 (Supplementary Fig. 5). There was also no overt increase in cytotoxicity as measured by LDH release with treatment (Supplementary Fig. 4i).

## Exogenously applied AD-derived Aβ binds human post-synaptic structures

We next sought to examine the impact of disease-associated Aβ on synapses using array tomography, qPCR and Western blot. Unlike many studies which use supraphysiological concentrations of synthetic Aβ, we sought to characterise the impact of pathophysiological levels of AD-derived Aβ on live adult human brain slice cultures. We used a protocol previously shown to generate AD-derived brain extract containing highly bioactive low molecular weight Aβ species[105]. We characterised the Aβ species present in our AD-derived brain extract using ELISA and showed the Aβ containing (Aβ+ve) brain extract has 0.6 pM $Aβ_{1-42}$ and 28.2 pM $Aβ_{1-40}$ (further characterisation available in Supplementary Fig. 6). We also generated an Aβ negative (Aβ-ve) AD-brain derived extract immunodepleted for Aβ (Aβ levels below detectable threshold). HBSCs from the same patient were split into three conditions in a repeated measures design. Cultures were then exposed to either medium only control, medium containing 25% soluble AD-brain extract (Aβ+ve) or medium containing 25% soluble AD-brain extract immunodepleted for Aβ (Aβ-ve). HBSCs were exposed to treatments for 72 hours before collection for array tomography, Western blot, and qPCR analysis. Array tomography (Fig. 5a) revealed that HBSCs exposed to the Aβ+ve treatment showed a significant increase in the volume of oligomeric Aβ staining in the tissue when compared to Aβ-ve treatment (Fig. 5b). Interestingly, whilst there was no change in the level of oligomeric Aβ co-localising with the pre-synaptic marker synaptophysin (Fig. 5c), there was an increase in Aβ co-localising with the post-synaptic marker PSD95 (Fig. 5d), indicating a preferential binding to post-synaptic compartments in human cortical tissue. In the Aβ+ve group there was a decrease in synaptophysin density (Fig. 5e) but PSD95 density remained unchanged compared to the Aβ-ve group (Fig. 5f). Raw data, not normalised to medium-only control, is available in (Supplementary Fig. 7). In a separate set of cases, there were no observed alterations in gene expression for *GAD2*, *GFAP*, *IBA1*, *SNAP25*, *SYP* or *SYT1* following either AD-brain extract treatment (Fig. 5g). Neither Aβ+ve or Aβ-ve treatment resulted in loss of neuronal proteins Tuj1 or PGP9.5 by Western blot (Supplementary Fig. 8), indicating that soluble AD-brain extract treatment did not result in an overall loss of neurons. We also did not find any changes in the levels of synaptophysin, Synapsin 1, PSD95, SAP97, or GFAP by Western blot

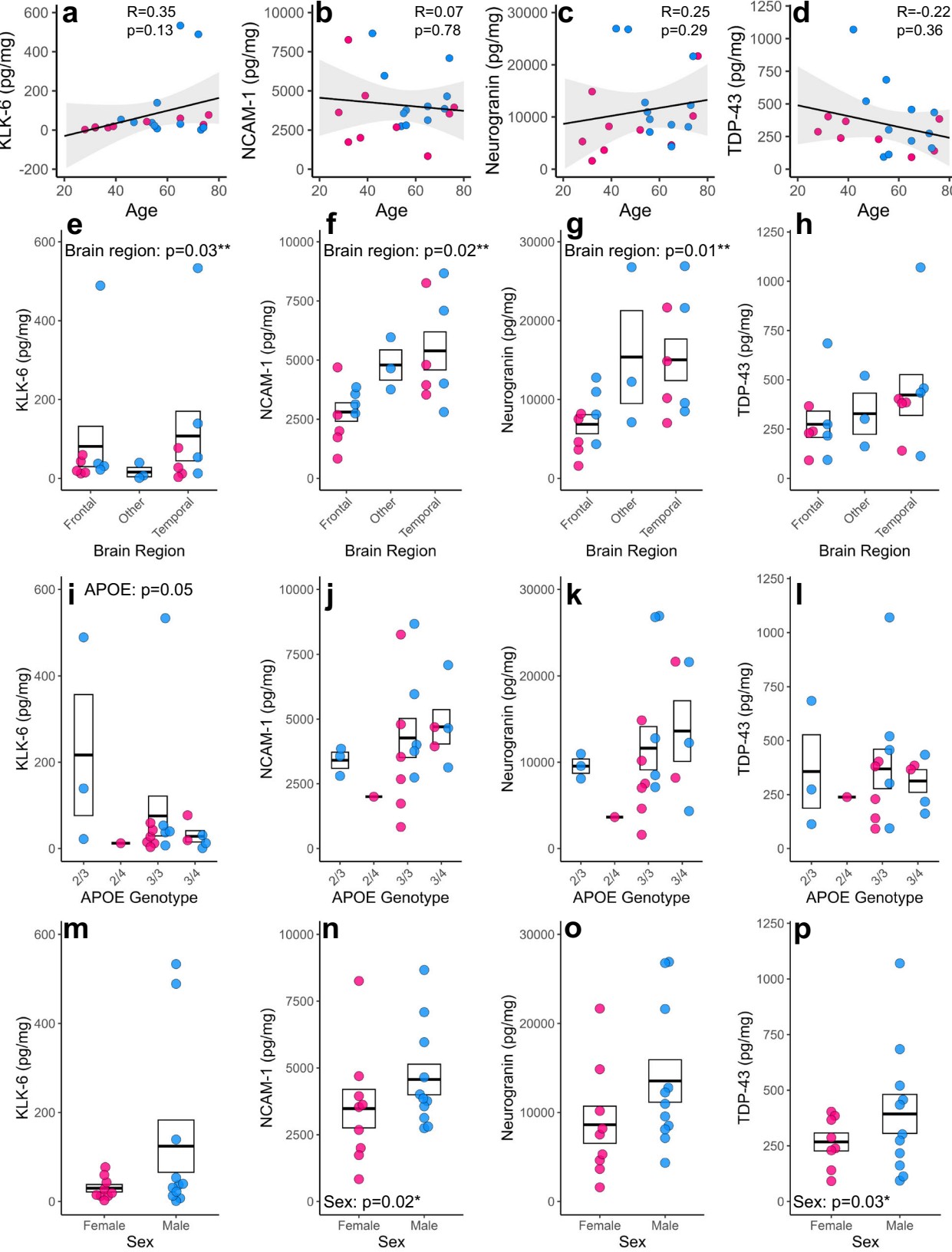

between the Aβ+ve or Aβ-ve treatment groups (Supplementary Fig. 8). However, there was a significant increase of PSD95 and SAP97 protein levels following soluble AD-brain extract treatment, regardless of Aβ status, compared to control, perhaps indicating a compensatory response or presence of potentially pro-synaptic factors within the extract (Supplementary Fig. 8).

**Aβ and tau pathological features are detected in live human brain samples from older patients, and are preserved in culture**
Finally, we screened our HBSC samples for evidence of spontaneously occurring AD-relevant neuropathology. Whilst none of our donors had been diagnosed with a neurodegenerative disorder, 1 in 3 individuals are expected to develop dementia over their lifetime[106], with

**Fig. 3 | Relationship between other neurodegenerative biomarkers and patient characteristics.** Proteins of interest in the culture medium were measured by a multiplex assay and normalised to total protein content in the medium. **a–d** Scatter plots showing the correlations between age of patient and KLK-6 ($r_{s(20)}$ = −0.35, $p$ = 0.13) (**a**), NCAM-1 ($r_{s(20)}$ = 0.07, $p$ = 0.78) (**b**) Neurogranin ($r_{s(20)}$ = 0.25, $p$ = 0.29) (**c**) and TDP-43 ($r_{s(20)}$ = −0.22, $p$ = 0.36) (**d**) detected in the culture medium. Regression lines are plotted on, with shaded area representing 95% confidence intervals. Statistics: Spearman's rank correlation due to non-normal distribution, $n$ = 20 human cases for above analysis except for TDP-43, where $n$ = 19. **e–h** Box and dot plots assessing the relationship between brain region and KLK-6 (Tukey, $\chi^2_{(2,20)}$ = 6.74, $p$ = 0.03*) (**e**) NCAM-1 (gaussian, $\chi^2_{(2,20)}$ = 8.3, $p$ = 0.02*) (**f**) Neurogranin (Tukey, $\chi^2_{(2,20)}$ = 8.73, $p$ = 0.01**) (**g**) and TDP-43 (gaussian, $\chi^2_{(2,20)}$ = 2.08, $p$ = 0.35) (**h**) in the culture medium. Statistics: GLMM with format variable - Brain region + Sex +(1|Age), $n$ = 20 human cases for above analysis except for TDP-43, where $n$ = 19. **i–l** Box and dot plots assessing the relationship between *APOE* genotype and KLK-6 (Tukey, $\chi^2_{(3,20)}$ = 7.8, $p$ = 0.05) (**i**), NCAM-1 (gaussian, $\chi^2_{(3,20)}$ = 2.0, $p$ = 0.57) (**j**), Neurogranin (Tukey, $\chi^2_{(3,20)}$ = 1.58, $p$ = 0.66) (**k**) and TDP-43 (gaussian, $\chi^2_{(3,20)}$ = 1.58, $p$ = 0.28) (**l**) detected in the culture medium. Statistics: GLMM with format variable - APOE + Sex + (1|Age), $n$ = 20 human cases for above analysis except for TDP-43, where $n$ = 19. **m–p** Box and dot plots assessing the relationship between sex and KLK-6 (Tukey, $\chi^2_{(1,20)}$ = 0.73, $p$ = 0.39) (**m**) NCAM-1 (log, $\chi^2_{(1,20)}$ = 5.7, $p$ = 0.02*) (**n**) Neurogranin (Tukey, $\chi^2_{(1,20)}$ = 3.25, $p$ = 0.07) (**o**) and TDP-43 (gaussian, $\chi^2_{(1,20)}$ = 4.73, $p$ = 0.03*) (**p**) detected in the culture medium. Statistics: GLMM with format variable - Sex + (1|Age), $n$ = 20 human cases for above analysis except for TDP-43, where $n$ = 19. **a–p** Pink dots are for female cases and blue dots for male cases. **e–p** Box represents SEM, with thick lines representing the mean. Source data are provided as a Source Data file.

pathological changes in the brain likely occurring decades before clinical disease presentation[107]. We therefore screened all cases used in this study (tissue availability limiting; $n$ = 35) for AD-associated pathology. Immediately following tissue slicing (acute), a subset of slices were fixed and immuno-stained for fibrillar-oligomeric Aβ (OC) and phosphorylated tau (phospho-tau Ser202/Thr205 (AT8)). Slices were then screened for pathological features and scored along the following criteria. For Aβ, we documented the presence or absence of extracellular Aβ plaques (Fig. 6a). For phospho-tau, we assessed whether we could observe neuropil thread-like staining or somatic tangle-like structures (Fig. 6d). Whilst we did detect Aβ staining in glial cells or blood vessels in a number of our samples, only 17% had extracellular OC staining typical of Aβ plaques (Fig. 6b). Phospho-tau staining revealed just under a third of the cohort have signs of tau pathology, with 20% having evidence of neurofibrillary tangles (Fig. 6e). We found a significant effect of donor age on the presence of Aβ plaques (Fig. 6c) and on pathological phospho-tau (Fig. 6f), with greater levels of pathology seen in older donors. We did not detect Aβ plaques or neurofibrillary tangles in patients below the age of 50 years old. Having established that we could detect Aβ plaques in acute neocortical access tissue, we next sought to explore whether such features were observable, and retained, over time in live HBSCs. We found that Aβ plaques could be labelled and live-imaged via multiphoton microscopy in HBSCs using methoxy-X04 (Fig. 6k) or Thioflavin-S (Fig. 6l) at 3 *div*. We were also able to observe Aβ plaques in proximity with MAP2 positive neurites and neuronal cell bodies in HBSCs fixed and immunostained after 7 *div* (Fig. 6m, n). We did not find a relationship between the brain region taken and the proportion of samples with either Aβ (Fig. 6g) or tau pathology (Fig. 6i). There was also no relationship observed between Aβ or tau pathology and patient *APOE* genotype, but we were limited by small group sizes for some *APOE* genotypes (Fig. 6h, j).

Following observations that pathological features were retained during the culture period, we then examined how the presence of Aβ or tau pathology in the slice impacted the concentration of Aβ$_{1-40}$, Aβ$_{1-42}$ or tau in the culture medium at 7 *div*. (Fig. 7). Whilst there was no relationship between plaque pathology and Aβ$_{1-40}$ (Fig. 7a), Aβ$_{1-42}$ (Fig. 7b) or total tau (Fig. 7d) concentration in the medium, there was a trend for increased Aβ$_{1-42}$/Aβ$_{1-40}$ ratio in samples with plaques (Fig. 7c). There was no relationship between plaque pathology and KLK-6, NCAM-1, Neurogranin or TDP-43 release into the medium. For tau pathology, there was similarly no effect of threads or tangles on Aβ$_{1-40}$ (Fig. 7g), Aβ$_{1-42}$ (Fig. 7h), total tau (Fig. 7j), NCAM-1, Neurogranin or TDP-43 (Fig. 7n–p) concentration in the medium, but there was a significant impact of pathology on the Aβ$_{1-42}$/Aβ$_{1-40}$ ratio (Fig. 7i) and KLK-6 release (Fig. 7m).

## Discussion

To our knowledge, this is the first study to evaluate endogenous Aβ and tau dynamics, and spontaneous pathology in adult human brain slice cultures as well as assessing their response to pharmacological agents and human AD-derived Aβ. When examining endogenous protein concentrations in the HBSC medium, there was no significant relationship between patient age and Aβ$_{1-42}$, Aβ$_{1-42}$/Aβ$_{1-40}$ ratio, tau, KLK-6, NCAM-1, Neurogranin or TDP-43 release into the medium. We did observe a trend towards reduced soluble Aβ$_{1-40}$ with increasing patient age. We believe the HBSC medium is most comparable to in vivo cerebral ISF[108]. Whilst some studies have examined Aβ and tau turnover in human CSF using stable isotope labelling kinetics studies[28–32,109], and direct sampling of ISF is regularly conducted in mice[110–113], very few studies, to our knowledge, have directly sampled ISF levels of Aβ, tau or other putative neurodegenerative biomarkers in living human patients. One study linked increased Aβ levels with improved neurological status after acute brain injury[43], whilst another found that ISF levels of Aβ$_{1-42}$ were lowest in Normal Pressure Hydrocephalus (NPH) patients with cortical Aβ pathology compared to those without[44]. Whilst the brain injury study did not look at patient age as a variable, it may be that the higher levels of Aβ$_{1-40}$ we see in HBSC medium from younger individuals reflects increased neuronal activity in these samples, either through baseline patient characteristics or tolerance to culture conditions. Interestingly, we did not find a correlation between the level of Aβ or tau with the levels of synaptophysin, PSD95, or neuronal proteins present in the slices at 7 *div*, indicating it is not a simple case of younger samples containing or retaining more neurons or synapses. In contrast to the NPH ISF study, we saw a trend towards increased Aβ$_{1-42}$/Aβ$_{1-40}$ in cases with plaque pathology compared to those without, with no effect on Aβ$_{1-40}$ or Aβ$_{1-42}$ levels. In addition, one case which had high spontaneous Aβ plaque burden in the tissue also had the highest levels of medium Aβ$_{1-42}$ of the entire cohort (-10-fold higher). Interestingly, we also found a significant effect of tau pathology on the Aβ$_{1-42}$/Aβ$_{1-40}$ ratio and levels of KLK-6 in the culture medium, with samples containing tangles showing the highest Aβ$_{1-42}$/Aβ$_{1-40}$ ratio and altered levels of KLK-6[114]. More commonly conducted studies examining CSF from human patients have observed an overall decline in Aβ$_{1-42}$ and Aβ$_{1-42}$/Aβ$_{1-40}$ ratio during normal ageing, which is exacerbated by both *APOE4* genotype and a diagnosis of AD[16–19,30,32,115]. The lack of *APOE* genotype effects on amyloid (although limited by small group sizes for some genotypes) and no change in Aβ$_{1-42}$ in our sample set differs from this, which may reflect different processes being captured in HBSC medium versus CSF/ISF or could be due to limits in ELISA sensitivity for Aβ$_{1-42}$. By contrast, our findings align well with studies in post-mortem brain tissue, that show the levels of soluble Aβ$_{1-40}$ decline with age[40,41]. Further studies examining brain tissue, HBSC medium, ISF, and CSF from the same patient donor could be highly informative in untangling how changes observed in HBSCs correlate with in vivo readouts.

Our finding that the levels of total tau in the HBSC medium were increased in temporal lobe samples when compared to frontal is of interest in the context of regional vulnerability to AD. Whilst four distinct patterns of tau spread have been reported in AD[96], pathology

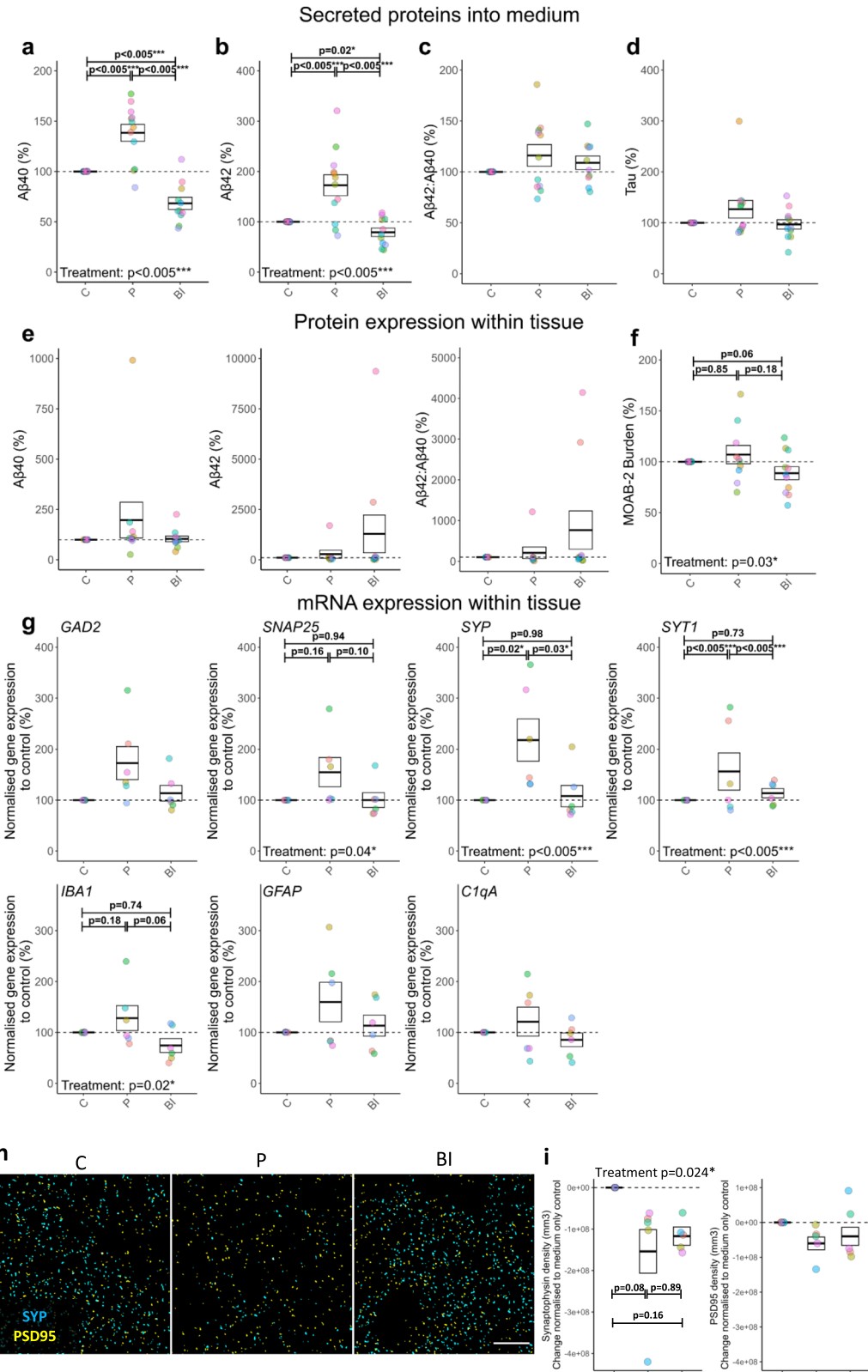

**Secreted proteins into medium**

**Protein expression within tissue**

**mRNA expression within tissue**

generally originates in the medial temporal lobe[94,95]. Post-mortem brain studies report that in healthy individuals, there is no difference in endogenous tau concentration between temporal and frontal lobes[116,117]. As we did not see an effect of age on tau concentration in the HBSC medium, we hypothesise there could be regional differences in tau release in human brain tissue. Increased neuronal activity increases tau release in mice[118], so it may be that there are baseline differences in

neuronal activity or tau release in this region that could contribute to greater vulnerability to tau pathology in later life. Interestingly, we found a similar effect of brain region on the medium levels of KLK-6, NCAM-1 and Neurogranin, with no significant alterations in TDP-43. The differing protein levels in the culture medium again contrast with lack of evidence for differential expression of these putative biomarkers within brain tissue (as reported by The Brain Resource of the

**Fig. 4 | BACE1 inhibition and Phosphoramidon application for 7 *div* bi-directionally modulates Aβ production, altering synaptic gene expression.**
**a–g** and **i** Box and dot plots showing normalised % change from control values when treated with Phosphoramidon or BACE1 inhibitor. Dots are coloured based on Case ID. Dashed line at 100% represents baseline (control), which samples were normalised to. Box represents SEM, thick line represents mean. Statistics: GLMM (variable - Drug treatment + Sex + (1|Age) + (1|Case ID)). Statistics performed on raw data (non-normalised; Supplementary Fig. 4c). C Control, P Phosphoramidon, BI BACE1 inhibitor. **a–d** Bidirectional secretion in response to Phosphoramidon and BACE1 inhibitor was seen with $A\beta_{1-40}$ (gamma, $\chi^2_{(2,36)} = 108.41$, $p = 2.2e{-}16$***) (**a**) and $A\beta_{1-42}$ (gamma, $\chi^2_{(2,36)} = 53.37$, $p = 2.6e{-}12$***) (**b**). No alterations were seen on $A\beta_{1-42}/A\beta_{1-40}$ ratio (gaussian, $\chi^2_{(2,36)} = 3.84$, $p = 0.15$) (**c**) and tau (log, $\chi^2_{(2,36)} = 2.91$, $p = 0.23$) (**d**) secretion. $N = 36$ measures from 13 cases; $n = 13$C, $n = 12$P, $n = 11$ BI. **e** $A\beta_{1-40}$ (Tukey, $\chi^2_{(2,30)} = 3.20$, $p = 0.20$), $A\beta_{1-42}$ (Tukey, $\chi^2_{(2,30)} = 0.73$, $p = 0.70$), and $A\beta_{1-42}/A\beta_{1-40}$ (Tukey, $\chi^2_{(2,30)} = 3.20$, $p = 0.20$) did not show significant differences in tissue protein expression in response to treatment. $n = 32$ measures from 11 cases; $n = 11$ C, $n = 10$ P, $n = 11$ BI. **f** MOAB-2 expression showed significant alterations to treatment (gaussian, $\chi^2_{(2,32)} = 6.73$, $p = 0.03$*), but did not show *post-hoc* significant differences. $n = 128$ measures from 11 cases; $n = 11$ C, $n = 10$ P, $n = 11$ BI. Values represent the average of four regions. **g** *SNAP25* (Tukey, $\chi^2_{(2,18)} = 6.37$, $p = 0.04$*), *SYP* (Tukey, $\chi^2_{(2,18)} = 14.01$, $p = 0.0009$***), *SYT1* (gamma, $\chi^2_{(2,18)} = 14.9$, $p = 0.0006$***), and *IBA1* (Tukey, $\chi^2_{(2,18)} = 7.60$, $p = 0.02$*) showed treatment effects. *GAD2* (gaussian, $\chi^2_{(2,18)} = 3.61$, $p = 0.16$), *GFAP* (Tukey, $\chi^2_{(2,18)} = 1.75$, $p = 0.42$), and *C1qA* (gaussian, $\chi^2_{(2,18)} = 2.72$, $p = 0.26$) did not show significant treatment effects. Gene expression normalised to the control treatment group. n = 6 cases. **h** and **i** Pre- (synaptophysin, cyan) and post- (PSD95, yellow) synaptic immunostaining from treated slices processed for array tomography. $N = 5$ cases. **h** A single 70 nm segmented section from each culture condition shown. Scale bar = 10 μm. **i** Treatment decreased synaptophysin density (cube root, $\chi^2_{(2,7.75)} = 7.44$, $p = 0.024$*), but not PSD95 density (Tukey, $\chi^2_{(2,7.75)} = 7.44$, $p = 0.33$). Data points refer to case mean, shown as subtraction normalised to medium control. Statistics: LMEM (synapse_density - treatment + 1|Case/Case_treatment). Between treatment group *p*-values were calculated from *post-hoc* testing with Tukey correction for multiple comparisons. Source data are provided as a Source Data file.

Human Protein Atlas[119–123]). Future exploration of region-specific protein release is uniquely suited to HBSCs, as regional ISF measurements would be exceptionally challenging in human patients, and CSF measurements do not provide spatial information. Understanding how the release of proteins differs between brain regions and associates with patient characteristics could provide mechanistic insight into the origin and progression of different neurodegenerative diseases.

HBSCs facilitate testing potential therapeutics in live adult human brain tissue, as well as assessing the impact of increased disease-related protein levels—an impossible experiment in human patients. Human trials have shown that BACE1 inhibitors reduce plaque load and lower CSF Aβ with little change to total or p-tau[64]. In line with this, we found that the application of BACE1 inhibitor LY2886721 to HBSCs resulted in a significant decrease in $A\beta_{1-40}$ and $A\beta_{1-42}$ in the HBSC medium with no change to tau levels or synaptic transcripts. Previous studies have shown that administration of Phosphoramidon to mice or cell lines results in around a 2-fold increase in Aβ levels[100–103,124]. For the first time in live human tissue, we find that Phosphoramidon increases endogenous levels of Aβ and, results in increased levels of synaptic transcripts. Interestingly, we found an overall effect of treatment on synaptophysin density, with both increasing (Phosphoramidon) and lowering (BACE1 inhibitor) Aβ concentration resulting in a loss of puncta, without evidence of cellular toxicity (no change of measured LDH release). The lack of change in synaptic proteins detected by Western blot could be explained by the reduced sensitivity of Western blot, or that alterations in puncta may be due to synaptic reorganisation. Research suggests that elevated Aβ damages synapses[1], but it's crucial to highlight that most prior studies explore supraphysiological levels of exogenous Aβ, not physiological increases in endogenous Aβ. Indeed, evidence from mouse studies points to a hormetic effect of Aβ, where high levels of Aβ are synaptotoxic, but picomolar increases in Aβ often enhance synaptic function[3,66,125–131]. Our observation that manipulation of Aβ concentration away from baseline in either direction results in loss of presynaptic puncta provides evidence that a similar hormetic role of Aβ may be present in human brain tissue. The fact that a small physiological increase in Aβ also coincides with an increase in synaptic transcripts could indicate a compensatory response or represent a period of synaptic reorganisation/repair, whilst the loss of Aβ through BACE inhibition is not compensated for and may therefore have longer-term consequences. Indeed, previous studies in wild-type mice treated with BACE1 inhibitors have reported spine loss, cognitive deficits, and alterations to LTP[132]. Future studies exploring the long-term impact of small fluctuations in Aβ concentration on synapse survival and function will be highly informative.

In contrast to enhancing endogenous production of Aβ in HBSCs, the application of exogenous AD-brain derived Aβ resulted in the loss of synaptic puncta without evidence of compensatory changes to synaptic transcripts. AD-brain derived Aβ species, even at low picomolar concentrations as used here, have previously been shown to be toxic to synapses[1,45–48]. We provide the first direct evidence that exogenously applied human-derived Aβ preferentially binds to post-synaptic structures in live human brain tissue, in agreement with prior work in rat hippocampal neurons[133] and rhesus macaque[49]. Whilst we find that oligomeric Aβ preferentially binds to post-synaptic structures, interestingly it is pre-synaptic puncta that are lost in our model. This builds on studies using free-floating human brain slices that demonstrated synthetic Aβ accumulates in brain tissue and results in a loss of synaptophysin mRNA[58,59]. In contrast with this study, we do not see any changes to synaptic mRNA in response to pathological Aβ exposure, suggesting that this more physiological insult may result in puncta loss without neuronal loss. This paper also did not look at synapse loss directly, so it is unclear if their observed loss of mRNA equated to loss of synaptic puncta as we report here. Whilst the loss of synaptophysin puncta we observe is modest, it equates to a mean loss of 102,528,403 synapses per mm³ which may have significant consequences for network function. Further exploration of electrophysiological consequences of Aβ-induced synaptophysin loss, as well as how pathological Aβ derived from human AD cases differs to synthetic Aβ, will be highly informative and help direct future studies to the best treatment type to accurately model AD-like changes. Other studies have previously shown that pre-synaptic proteins are lost first in response to amyloid pathology[58,59,134,135], so it may be that despite Aβ being located post-synaptically, pre-synapses are more vulnerable to downstream pathological consequences. Studies in human post-mortem brain tissue have demonstrated that synaptic oligomeric Aβ and synapse loss are most prominent in the vicinity of Aβ plaques and accumulate to a greater degree in *APOE* ε4 carriers[55–57], so future work exploring the impact of patient characteristics in response to Aβ challenge in HBSCs may help provide mechanistic insight into the relationship between *APOE* genotype and AD risk.

HBSCs offer valuable insights into brain physiology and AD pathology, but we must be aware of their limitations. The tissue is from brain cancer patients, and while peritumoral samples are often similar to epilepsy tissue[69,79], tumour-related cellular changes and exposure to treatments may affect results. As HBSCs are cultured resected tissue, they may show altered protein expression over time in culture, including loss of some synaptic proteins, which should be considered when comparing to in vivo conditions. Additionally, patient demographics (age, genotype, sex) introduce variability, and we are underpowered for certain *APOE* genotypes. To mitigate these factors, we control for variability using a repeated measures design, normalising responses to untreated controls, as well as controlling for and testing

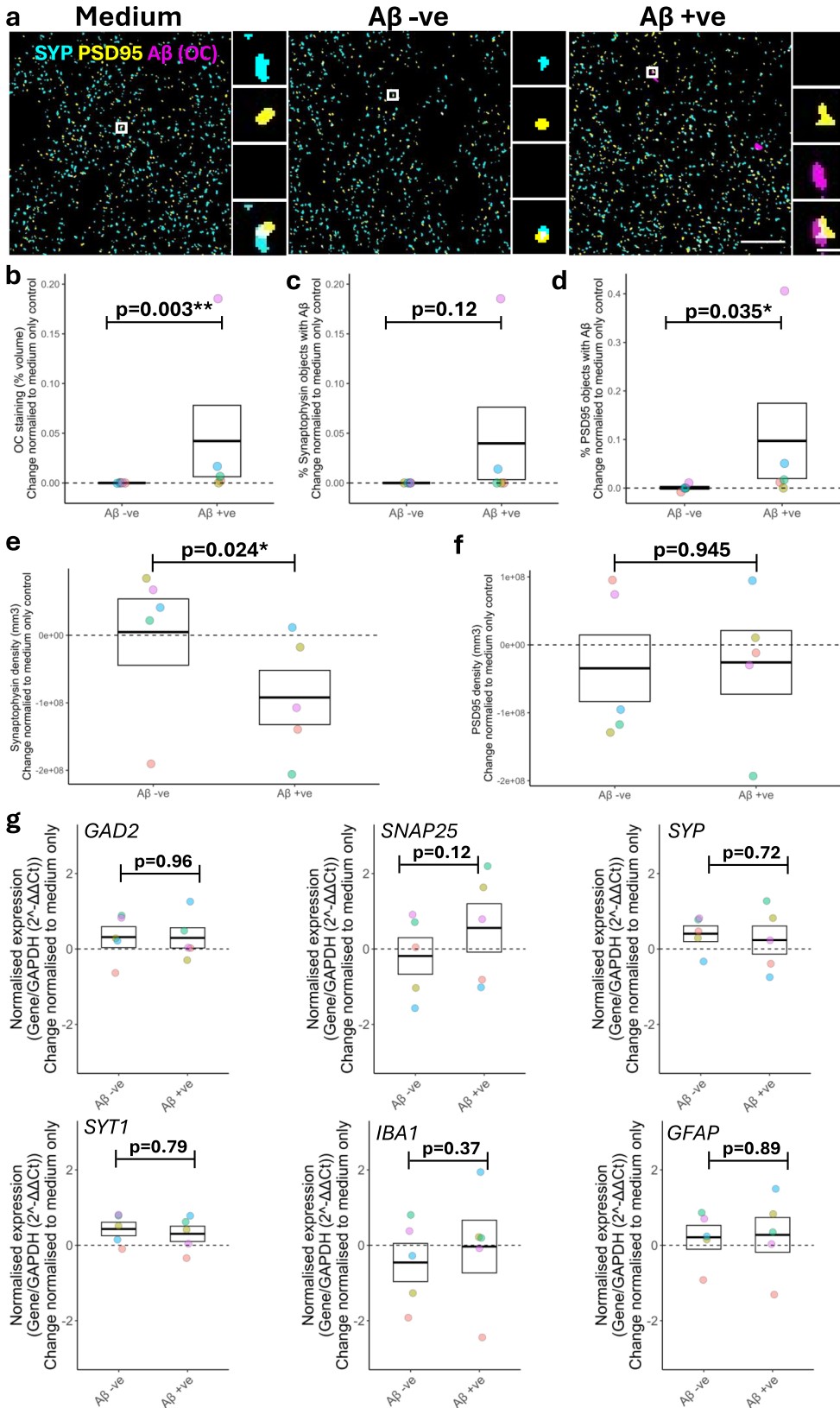

for effects of known variables where possible. In our study, we find very few significant interactions with sex, with the exception of an increase in the levels of NCAM-1 and TDP-43 released into culture medium in samples derived from male patients. Importantly, none of the patients have been diagnosed with AD, and long-term follow-up is sadly limited due to the poor prognosis of brain cancer patients. Widening our

sample pool in the future to include patients with epilepsy, or NPH biopsies (both of which show much greater long-term survival) could open doors to long-term follow-up studies.

The fact that we can detect Aβ and tau pathology in a subset of our patient samples, and that these features are retained and can be live-imaged in culture does, however, open exciting future avenues to

**Fig. 5 | AD-derived Aβ colocalises to post-synapses and induces loss of synaptophysin after 72 hours in human brain slice cultures. a** Single 70 nm segmented array-tomography sections show representative staining and colocalisation in each culture condition. Pre-synapses (synaptophysin, SYP, cyan), post-synapses (PSD95, yellow), Aβ (OC, magenta). Scale bar in large box = 10 μm. Large boxes = 50 μm * 50 μm. Scale bar in small box = 1 μm. Small boxes = 2 μm * 2 μm. **b–g** Dots are coloured based on Case ID. *n* = 5 human cases. Dashed line at 0 represents baseline (control), which samples were normalised to. Box represents SEM, with the thick line representing the mean. Raw data shown in Supplementary Fig. 7. **b–d** Slices cultured with Aβ+ve soluble AD-brain extract show an increase in Aβ immunostaining (*β* = 0.56, 95% CI [0.186, 0.934], *z.ratio* = 2.936, *p* = 0.003***) (**b**) no change in the percentage of pre-synapses containing Aβ (*β* = 0.182, 95% CI [−0.046, 0.41], *z.ratio* = 1.563, *p* = 0.118) (**c**) and an increase in the percentage of post-synapses with Aβ (*β* = 0.472, 95% CI [0.034, 0.911], *z.ratio* = 2.11, *p* = 0.0348*) (**d**). Statistics: to ensure the assumptions of a statistical test were met, data was transformed into a binary readout. An increase in pathology was defined as a >2-fold increase in staining when compared to medium-treated control. Statistics: GLMM (variable ~ Treatment + (1|Case), family = binomial). **e** and **f** Culturing slices with Aβ+ve soluble AD-brain extract induced a loss of synaptophysin puncta (*β* = 501, 95% CI [89.8, 912], *t*(6.64) = 2.913, *p* = 0.02*) (**e**). There was no effect on PSD95 density (*β* = 18.1, 95% CI [−573, 609], *t*(7.86) = 0.071, *p* = 0.945) (**f**). Statistics: Linear mixed effects model (LMEM) (variable ~ Treatment + (1|Case)). **g** Treatment showed no change in mRNA expression of *GAD2* (*t*(52) = 0.049, *p* = 0.96), *SNAP25* (*t*(52) = −1.591, *p* = 0.12), *SYP* (*t*(52) = 0.362, *p* = 0.72), *SYT1* (*t*(52) = 0.273, *p* = 0.79), *IBA1* (*t*(52) = −0.897, *p* = 0.37) or *GFAP* (*t*(52) = −0.135., *p* = 0.89). Statistics: LMEM (gene ~ Treatment*Gene + (1|Case)). *p*-values calculated from *post-hoc* testing with Tukey correction for multiple comparisons. *N* = 5 human cases. Source data are provided as a Source Data file.

explore potential sporadic pre-clinical AD pathology in the live adult human brain. Whilst we cannot know for sure that the pathology detected in the samples would go on to cause symptomatic disease in that patient, we are able to explore how the presence of these early features impacts other biological readouts. Whilst there is some evidence that cetaceans, non-human primates, dogs and cats can develop spontaneous AD-like pathology[136–138], wildtype mice do not develop pathology without the introduction of rare familial-AD mutations, often alongside overexpression of humanised APP or tau[139]. Compared with rodents, the human brain also contains seven times as many neurons and double the amount of synapses[11–14]. It seems clear then, especially for diseases such as Alzheimer's disease, that increased use of human model systems will improve our chances of translational success. Indeed, our previous work using HBSCs demonstrates that responses to pharmacological agents can differ between mouse and human brain slice cultures[87]. We also recently showed that oligomeric tau derived from the brains of individuals with Progressive Supranuclear Palsy (PSP) can be taken up into synapses, and induce astrogliosis in HBSCs[88]. By correlating responses to treatments, as well as changes in endogenous dynamics, to patient age, sex, brain region, genetics and lifestyle risk factors, we propose that, despite some limitations, HBSCs represent an extremely powerful tool for both discovery and translational research.

## Methods

### Ethical approval and patient characteristics
Ethical approval for the use of surplus neocortical access tissue originating from patients undergoing tumour resection surgery was provided by the Lothian NRS Bioresource (REC number: 15/ES/0094, IRAS number: 165488, Bioresource No SR1319). NHS Lothian Caldicott Guardian Approval (Approval number CRD19080) was obtained to receive data on the exact patient age (in years), sex, brain region provided and reason for surgery. Patients provided informed consent to donate surplus tissue (which would normally be discarded during surgery) and permit genetic testing of tissue, by signing the Lothian NRS Bioresource Consent form after discussion with a research nurse or neurosurgeon. Patient details are listed in Table 1, with ages reported to the nearest 5 years to enhance patient confidentiality. All data analysis in this study has been performed using the exact patient age to the nearest year. Patient sex was determined from medical records. For the generation of AD-brain soluble extract, post-mortem human temporal and frontal cortex tissue from people who died of Alzheimer's disease was acquired from the Edinburgh Brain Bank and the Massachusetts Alzheimer's Disease Research Center (grant 1P30AG062421-01). Informed consent for human post-mortem brain tissue collection was obtained from brain tissue donors before death and their next of kin post-mortem. Tissue was provided by the brain bank to the authors pseudo-anonymised without identifying details of the donor. The Edinburgh Brain Bank is a Medical Research Council-funded facility with research ethics committee approval (16/ES/0084).

Experiments were approved by the Edinburgh Brain Bank ethics committee, the Academic and Clinical Central Office for Research and Development (ACCORD) and the medical research ethics committee AMREC (a joint office of the University of Edinburgh and National Health Service Lothian), approval number 15-HV-016. The details of the post-mortem human cases are found in Supplementary Table 1 (ages reported to the nearest 5 years to enhance patient confidentiality). Inclusion criteria for cases included a clinical diagnosis of dementia and a neuropathological diagnosis of Alzheimer's disease with Braak stage V or VI. Exclusion criteria included substantial co-pathologies in the brain.

### Generation and maintenance of human brain slice cultures (HBSCs)
Dissection and culture methods to generate human brain slice cultures (HBSCs) were adapted from published studies[67–69,87,91]. A summary schematic of tissue dissection is shown in Fig. 1a. Briefly, surplus, non-tumour, neocortical access tissue, was excised from patients undergoing brain tumour debulking surgery. This tissue was then immediately placed in ice-cold oxygenated artificial cerebrospinal fluid (aCSF) containing: 87 mM NaCl, 2.5 mM KCl, 10 mM HEPES, 1.62 mM $NaH_2PO_4$, 25 mM D-glucose, 75 mM sucrose, 1 mM Na-Pyruvate, 1 mM ascorbic acid, 7 mM $MgCl_2$ and 0.5 mM $CaCl_2$. aCSF is at ~330 mOsm, is adjusted to a pH of 7.4 and is sterilised by passing through a 0.22 μm filter before use. Thick sections of pia are removed from the excised tissue using ultra-fine forceps and the block is trimmed to permit optimal orientation to preserve cortical layering. Tissue is embedded in 2% agar, set briefly on ice, then mounted onto a Leica VT1200S vibratome specimen plate using Loctite superglue. Tissue was cut at <0.1 mm/s speed in ice-cold oxygenated aCSF to generate 300 μm-thick slices. Slices were further sub-dissected using a scalpel, to generate ~3 mm wide columns containing all six cortical layers, and a small section of white matter. Once all sections were cut, slices were placed into a sterile wash solution (Wash Solution 1) composed of continuously oxygenated Hanks Balanced Salt Solution (HBSS, ThermoFisher: 14025092), HEPES (20 mM) and 1X penicillin–streptomycin (ThermoFisher: 15140122), which is ~ 305 mOsm and is adjusted to pH 7.3, for 20 minutes at room temperature. Slices were then plated on 0.4 μm pore culture membranes (Millipore: PICM0RG50) placed inside 35 mm sterile culture dishes, containing 750 μl of Wash Solution 2. Wash Solution 2 was composed of 96% BrainPhys Neuronal Medium (StemCell Technologies: 5790), 1X N2 (ThermoFisher: 17502001), 1X B27 (ThermoFisher: 17504044), 40 ng/ml hBDNF (StemCell Technologies: 78005), 30 ng/ml hGDNF (StemCell Technologies: 78058), 30 ng/ml Wnt7a (Abcam: ab116171), 2 μM ascorbic acid, 1 mM dibutyryl cAMP (APExBIO: B9001), 1 ug/ml laminin (APExBIO: A1023), 1X penicillin/streptomycin (ThermoFisher: 15140122), 3 units/ ml nystatin (Merck: N1638) and 20 mM HEPES. For all samples, the time taken from surgical tissue removal to slices being plated in Wash Solution 2 was kept to a minimum (<2 h). Slice

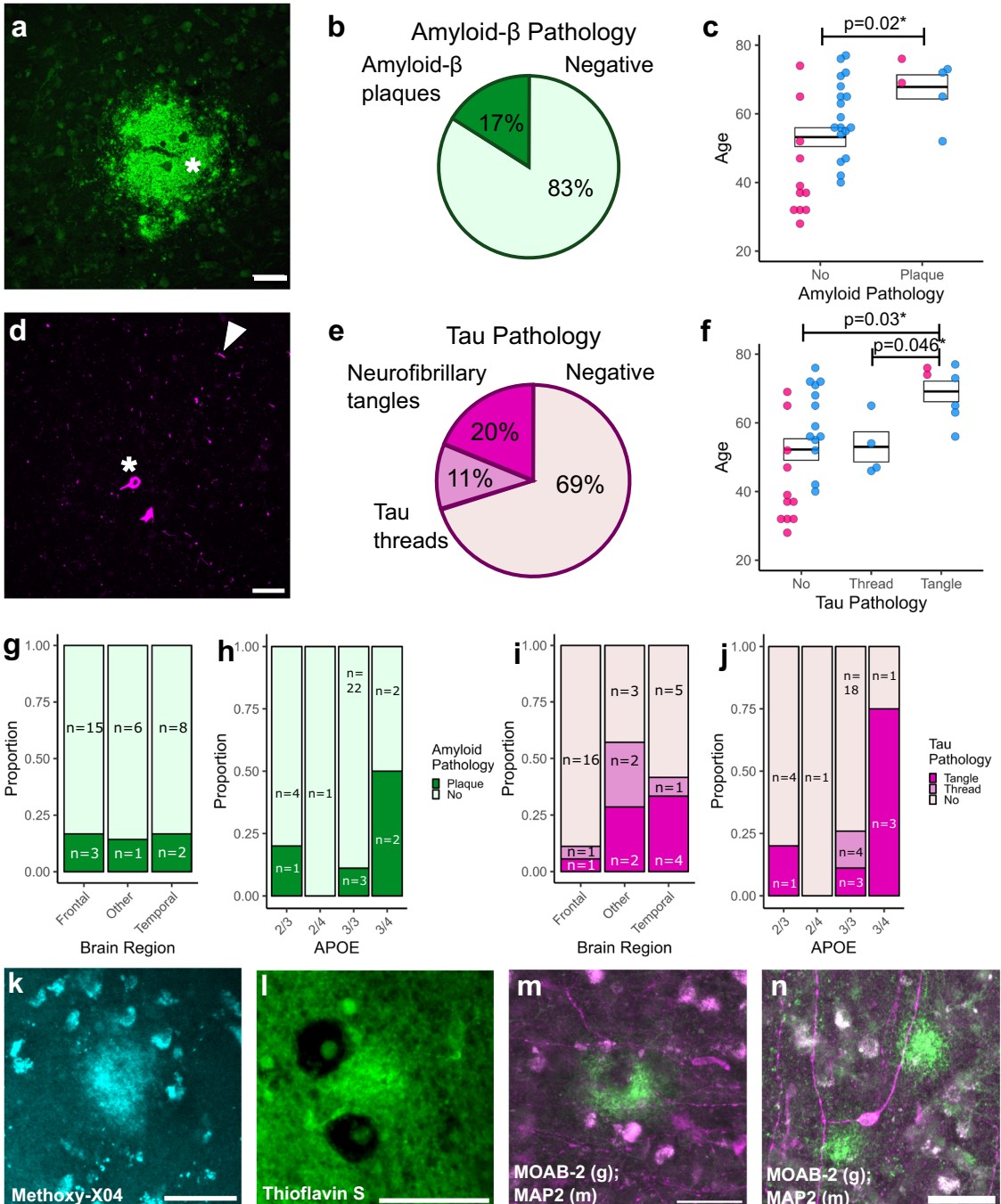

**Fig. 6 | Endogenous AD-related pathology visible in human brain slices. a** and **d** High magnification examples to highlight positive Aβ and tau pathology. **a** The asterisk denotes example Aβ plaques. **d** The asterisk highlights neurofibrillary tangle pathology, and the arrowhead highlights thread pathology. **b** and **e** Pie charts highlighting the percentage of cases that have Aβ plaque pathology, determined by visual inspection based on OC positive staining (**b**) or tau pathology, determined by visual inspection based on AT8 staining (**e**). **c** and **f** Box and dot plots showing the age of the patient compared to the Aβ (gaussian, $F_{(1,34)} = 5.99$, $p = 0.02^*$) (**c**) or tau (gaussian, $F_{(2,34)} = 4.6$, $p = 0.02^*$) (**f**) pathology score. A significant relationship between pathology score and age is seen for Aβ and tau pathology. Box represents the SEM, with the thick line representing the mean,

$n = 35$ human cases. **g–j** There was no relationship between Aβ pathology and brain region ($\chi^2_{(2,34)} = 0.1$, $p = 0.95$) (**g**), or *APOE* genotype ($\chi^2_{(3,23)} = 3.74$, $p = 0.29$) (**h**). There was no relationship between tau pathology and brain region ($\chi^2_{(4,34)} = 8.76$, $p = 0.07$) (**i**), or *APOE* genotype ($\chi^2_{(6,34)} = 10.04$, $p = 0.12$) (**j**), $n = 35$ human cases. **k** and **l** Live plaque imaging in culture using Methoxy-X04 or Thioflavin S. $N = 3$ human cases tested. **m** and **n** Immunofluorescent staining following culturing using MOAB-2 for Aβ pathology (green) and MAP2 for neurons (magenta), showing pathology is stable in culture for up to 7 *div*. $N = 3$ human cases tested. Scale bars are 50 μm. Statistics: Linear model (LM) with format variable - Pathology + Sex or chi-squared test. Source data are provided as a Source Data file.

cultures were kept in Wash Solution 2 in an incubator at 37 °C with 5% $CO_2$ for between 1 and 5 h. Following this, Wash Solution 2 was aspirated and replaced with pre-warmed sterile Maintenance Medium (identical composition to Wash Solution 2, but with HEPES removed).

After plating, 100% Maintenance Medium replacement occurred twice weekly at 4 *div* (days in vitro) and 7 *div* (unless otherwise specified). Culture Medium was collected and flash-frozen on dry ice at 7 *div* for further analysis.

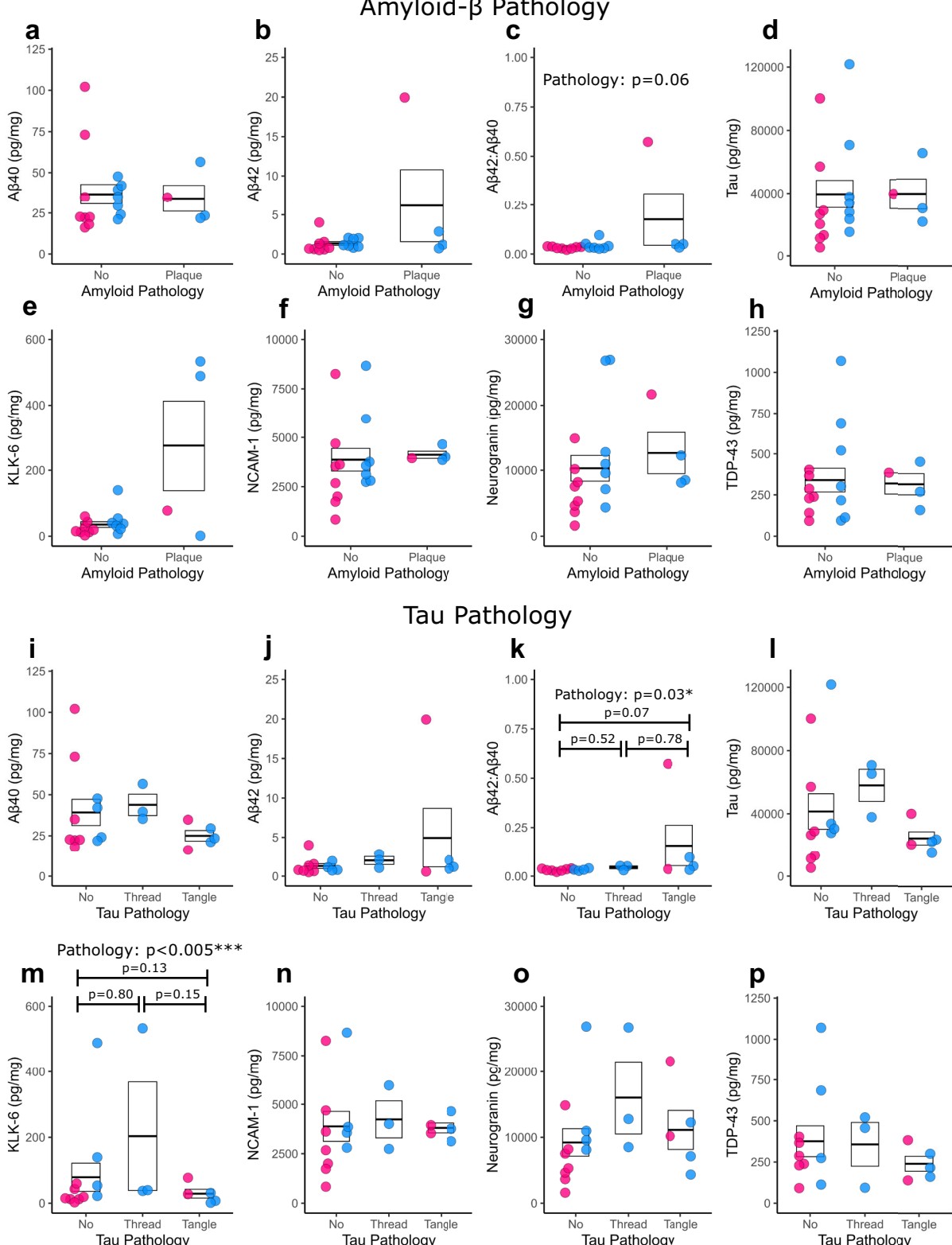

## General study design

To ensure reproducibility, all experiments were replicated in independent biological samples. As each human brain tissue case was collected, processed into slices and underwent treatments/imaging/analysis independently (due to random dates on which tissue was available) we consider each human case to be an independent repeat

of each experiment. For ELISAs, BCAs, and Luminex® Multiplex Assays, samples were stored in the freezer, and then run en mass with technical replicates to ensure validity. Positive controls and standard curves were used to ensure the experiments/assays had been successful. All microscopy/imaging experiments were conducted with the researcher blind to experimental conditions. As experimental readouts

**Fig. 7 | Assessment of the relationship between Aβ or tau pathology in the slice compared with protein release in the culture medium.** Aβ (**a**–**h**) and tau (**i**–**p**) pathology scoring (based on OC and AT8 immunostaining respectively) compared to Aβ$_{1-40}$ (**a** and **i**), Aβ$_{1-42}$ (**b** and **j**), Aβ$_{1-42}$/Aβ$_{1-40}$ (**c** and **k**), tau (**d** and **l**), KLK-6 (**e** and **m**), NCAM-1 (**f** and **n**), Neurogranin (**g** and **o**) and TDP-43 (**h** and **p**). For Aβ pathology (**a**–**h**), there was no significant effect on Aβ$_{1-40}$ (log, $\chi^2_{(2,19)} = 1.57$, $p = 0.21$) (**a**), Aβ$_{1-42}$ (Tukey, $\chi^2_{(2,19)} = 1.67$, $p = 0.20$) (**b**) tau (log, $\chi^2_{(1,19)} = 0.07$, $p = 0.80$) (**d**) KLK-6 (Tukey, $\chi^2_{(1,19)} = 0.97$, $p = 0.33$) (**e**), NCAM-1 (log, $\chi^2_{(1,19)} = 0.21$, $p = 0.65$) (**f**) Neurogranin (Gaussian, $\chi^2_{(1,19)} = 0.12$, $p = 0.73$) (**g**) or TDP-43 (Gaussian, $\chi^2_{(1,19)} = 0.02$, $p = 0.88$) (**h**) release into the medium. Aβ$_{1-42}$/Aβ$_{1-40}$ (Tukey, $\chi^2_{(1,19)} = 3.64$, $p = 0.06$) (**c**) showed a trend towards plaque cases having higher ratios. For tau pathology (**i**–**p**), there was no significant effect on Aβ$_{1-40}$ (log, $\chi^2_{(1,19)} = 4.84$, $p = 0.09$) (**i**), Aβ$_{1-42}$ (Tukey, $\chi^2_{(1,19)} = 3.84$, $p = 0.15$) (**j**), tau (Tukey, $\chi^2_{(1,19)} = 1.90$, $p = 0.39$) (**l**), NCAM-1 (log, $\chi^2_{(1,19)} = 0.1$, $p = 0.95$) (**n**), Neurogranin (Gaussian, $\chi^2_{(1,19)} = 1.1$, $p = 0.59$) (**o**) or TDP-43 (Gaussian, $\chi^2_{(1,19)} = 1.6$, $p = 0.45$) (**p**) release into the medium. There was a significant effect of tau pathology on Aβ$_{1-42}$/Aβ$_{1-40}$ ratio (Tukey, $\chi^2_{(2,19)} = 7.27$, $p = 0.027*$) (**k**) and KLK-6 (Tukey, $\chi^2_{(1,19)} = 15.9$, $p = 0.0004***$) (**m**) release into the medium. **a**–**p** Pink dots are for female cases and blue dots are for male cases. Box represents the SEM, with thick lines representing the mean. Statistics: GLMM with format variable - Pathology + Sex + (1|Age), $n = 19$ human cases, except TDP-43 $n = 18$. Source data are provided as a Source Data file.

## HBSC drug treatments

For the application of Aβ-modulating pharmaceuticals, 5 mM BACE1 inhibitor LY2886721 (Abcam: ab223886) and 100 mM Phosphoramidon (Sigma-Aldrich: R7385) stock solutions were generated by dissolving drugs in DMSO (Sigma-Aldrich: D2438) and sterile Milli-Q water respectively. Three treatment groups were allocated per human case, with a control condition having no treatment (i.e. 100% Maintenance Medium only). The respective drug treatments were added at 1/1000 dilution into Maintenance Medium to produce a final concentration of 5 μM BACE inhibitor or 100 μM of Phosphoramidon. 2 μl of medium (either control, BACE-1 inhibitor or Phosphoramidon treated) was added on top of each slice to ensure complete penetration of treatments through the slice tissue. Treatment was applied for 7 *div*, following which medium and slices were collected and processed as required. Doses were chosen in line with previous literature reports showing BACE1 inhibitor reducing the production of Aβ in a mouse slice culture[140] and Phosphoramidon effectively preventing the breakdown of Aβ in vitro[100].

## AD-brain soluble extract preparation

Soluble extracts from AD frontal and temporal cortex from Alzheimer's patients detailed in Supplementary Table 1 were generated as described previously[105]. Frozen samples were thawed on ice, finely diced and homogenised in aCSF (124 mM NaCl, 2.8 mM KCl, 1.25 mM NaH$_2$PO$_4$, 26 mM NaHCO$_3$), pH 7.4, supplemented with Complete Mini EDTA-free protease inhibitor cocktail tablets (Roche, 11836170001), using a Dounce homogeniser. Homogenised tissue was then transferred to 15 mL protein LoBind® tubes (Eppendorf, 0030122216), placed on a roller for 30 min at 4 °C, and then centrifuged at 2000×$g$ at 4 °C for 10 min to remove insoluble debris. The supernatant was removed and ultracentrifuged at 4 °C for 110 min at 200,000×$g$. The resulting supernatant, the soluble protein extract (s-extract), was dialysed in aCSF (as above) at 4 °C for 72 hours (Slide-A-Lyzer G2 Dialysis Cassettes) to remove salts, impurities and any pharmacological agents from the donor may have been exposed to. The aCSF was replaced every 24 hours. Following dialysis, the s-extract was pooled and divided in two for immunodepletion of total Aβ. The two portions of s-extract were incubated overnight at 4 °C with Protein A agarose (PrA) beads (30 μl/ml of sample) (ThermoFisher Scientific, 20334) with either anti-β-amyloid 17–24 (4G8, mouse IgG2b, 1:100, BioLegend, #SIG-39200) and anti-β-amyloid 1–16 (6E10, mouse IgG1 1:100 BioLegend, #SIG-39320) to generate "Aβ negative/Aβ -ve" s-extract, or mock-immunodepleted with an anti-GFP antibody (mouse IgG, 1:50, DSHB, DSHB-GFP-8H11) to create the "Aβ positive/Aβ +ve" s-extract. The s-extracts were then centrifuged at 2500×$g$ for 5 min, the supernatant was collected into fresh protein LoBind® tubes and the process was repeated two times (to result in a total of three overnight immunodepletion steps, each time the s-extract being exposed to fresh antibody and bead solutions). Finally, the Aβ negative and Aβ positive s-extract supernatants were collected into fresh protein LoBind® Eppendorf's and stored at −70 °C until needed. The concentration of Aβ$_{1-40}$ and Aβ$_{1-42}$ in the s-extracts was determined by commercially available ELISA kits (ThermoFisher Scientific: KHB3481 and KHB3544). Levels of oligomeric Aβ were assessed using a commercially available ELISA kit (Biosensis: #BSENBEK-2215).

## Immunogold electron microscopy

To further examine the species of Aβ present in the AD brain homogenate, 5 μl of homogenate was pipetted onto formvar-coated nickel grids (Agar Scientific, S162N1) and allowed to bind for 1 min, before removing the excess sample with filter paper. Grids were incubated in 1% sodium borohydride in 0.1 M PB for 5 min to reduce free aldehydes and thereafter incubated in 50 mM glycine in 0.1 M PB, for 10 minutes to inactivate residual aldehyde groups. The grids were then blocked in 0.1% BSA, 0.1% fish skin gelatin and 0.05% Tween in 0.2 M PB, pH 7.4, for 1 hour at room temperature. Anti- Amyloid-β antibody [MOAB-2] (Merck, MABN254, lot: 3713390) was diluted 1:25 in antibody dilution buffer (0.1% BSA, 150 mM sodium chloride in 0.2 M PB, pH7.4) and incubated with the nickel grids overnight at 4 °C, the grids were washed 6 times in 0.2 M PB, goat anti-mouse 10 nm gold-conjugated secondary antibody (Abcam, ab39619, lot: GR3379521-1) was diluted 1:50 in antibody dilution buffer and incubated with the nickel grids for 90 min at room temperature. The grids were then washed 6 times in 0.2 M PB, fixed in 2.5% glutaraldehyde in 0.2 M PB for 15 min, washed 6 times in 0.2 M PB, and finally rinsed with ddH$_2$O three times for 5 min each wash. Negative staining was carried out using 3% uranyl acetate in 50% ethanol for 15 min and 3% lead citrate in ddH$_2$O for 150 s. Images were captured using a transmission electron microscope (JEM-1400 Plus, JOEL).

## HBSC AD-brain soluble extract treatment

HBSCs from the same patient were split into three treatment groups in a repeated measures design. Group 1 was treated with 100% medium (control), Group 2 was treated with 75% medium: 25% Aβ negative s-extract (described above) (Aβ -ve), Group 3 was treated with 75% medium: 25% Aβ positive s-extract (described above) (Aβ +ve). 2 μl of treated culture medium was added on top of each slice to ensure complete exposure to treatments. Slices were collected after 3 *div* and either collected for western blot or fixed for array tomography processing.

## Array tomography

Slices were processed for array tomography as previously described[88,141]. Briefly, following collection, slices were fixed in 4% paraformaldehyde for 1.5 hours, dehydrated in ethanol and incubated in LR White resin (Agar Scientific: AGR1281) overnight at 4 °C. Individual samples were cured in LR White in gelatin capsules overnight at 53 °C. Samples were cut into ribbons of ultrathin serial 70 nm sections using a histo jumbo diamond knife (DiATOME, Agar Scientific) and an ultracut (Leica EM UC7). Ribbons were mounted on glass coverslips

(treated with fish-skin gelatin), dried on a slide warmer and outlined with a hydrophobic pen. Ribbons were rehydrated in 50 mM glycine in TBS and blocked in a solution of 0.1% fish skin gelatin and 0.05% Tween in TBS for 45 min at room temperature. Ribbons were incubated with primary antibodies (see Supplementary Table 2) at room temperature for 2 hours. Ribbons were washed with TBS, and secondary antibodies (see Supplementary Table 2) were applied for 45 min at room temperature. Ribbons were then washed with TBS. Finally, ribbons were mounted on glass slides with ImmuMount and imaged within 72 hours on a Zeiss Axio imager Z2.

## Microscopy and image analysis for array tomography

For array tomography imaging (Fig. 4), experimenters were blinded to case information during image processing and analysis. Images were acquired using Zen software using a ×63 oil immersion objective (1.4 numerical aperture) on a Zeiss AxioImager Z2 with a CoolSnap digital camera. Two regions of interest (ROIs) were chosen per slice and imaged in the same location on 15–30 sequential sections. Imaging parameters were kept the same, and an image was taken of the negative control in each session to ensure there was no non-specific staining. Individual images from each ROI were combined into a 3D image z-stack and a median background filter was applied in Image J with custom batch macros. Using custom MATLAB scripts, image stacks were aligned using rigid registration. Each channel was segmented using an auto-local thresholding algorithm to binarize images and remove any objects present in only a single section (noise). Segmented images were run through custom MATLAB and Python scripts to determine object density, colocalization between channels, and the burden of the staining (% volume of 3D image stack occupied by the stain). Objects were considered colocalized if at least 25% of the 3D volume overlapped. For pre- and post-synaptic objects to be considered a synaptic pair, the distance between the centre of each object had to be ≤0.5 μm. Saturation was minimised during image acquisition and only applied for figure visualisation. All custom software scripts are available on GitHub https://github.com/Spires-Jones-Lab. For more details, please see our methods video demonstrating this technique at https://doi.org/10.7488/ds/297.

## Immunofluorescence

For standard immunofluorescence techniques (Figs. 1 and 5), following culturing, slices were transferred to 4% PFA in PBS for between 2-12 hours and then stored in PBS until needed. Slices were rinsed in PBS with 1% Triton-X for 10 min. For Iba1, NeuN and MAP2, slices underwent an additional antigen retrieval step (10 min in a pressure cooker in IHC Antigen Retrieval Solution (Invitrogen: 00-4956-58), followed by cooling in ice for 30 min) before commencing the following protocol. Slices were transferred into 70% ethanol for 5 min. Slices were placed into autofluorescence eliminator reagent (Chemicon: 2160) for 5 min. Slices were then transferred into 70% ethanol for 5 min. All the following steps were performed on a shaker. Slices were rinsed 3 times for 10 min in PBS with 1% Triton-X, followed by a 1 hour block in normal goat/donkey serum (1:200, diluted in PBS with 1% Triton-x). Slices were then incubated in the required primary antibodies (details listed in Supplementary Table 3) overnight at 4 °C. Following this, slices were rinsed in PBS with 1% Triton-x 3 times. Slices were then incubated in the corresponding secondary antibody (details listed in Supplementary Table 3) at a concentration of 1:1000 in PBS with 1% Triton-x overnight at 4 °C. Slices were finally rinsed 3 times for 10 min in PBS. Slices were then mounted with fluoroprotectant mountant medium (Vectashield: H-1000) and imaged.

## Microscopy and image analysis for immunofluorescence

Slices were imaged on a multiphoton microscope (Leica TCS SP8; Coherent Chameleon laser at 760 nm), using LAS X software, under a ×25 objective. For pathology assessment (Fig. 5) whole slices were imaged and assessed for Aβ or tau pathology blinded to experimental condition of patient details. Immediately following tissue slicing, a subset of slices were fixed and immuno-stained for fibrillar-oligomeric Aβ (OC) and phosphorylated tau (phospho-tau Ser202/Thr205 (AT8)) (Fig. 5a). Slices were then screened for pathological features and scored along the following criteria. For Aβ, we documented the presence or absence of extracellular Aβ plaques (Fig. 5b). For phospho-tau, we assessed whether we could observe neuropil thread-like staining or somatic tangle-like structures (characteristic flame-shaped AT8-positive inclusions) (Fig. 5c). For assessing total Aβ burdens within the slice (using MOAB-2, a pan-Aβ that does not detect APP) (Fig. 3), a 50 μm stack from cortical layers I/II, III/IV, V/VI and white matter was taken in regions free from Aβ plaques. Each channel was segmented using an auto-local thresholding algorithm to binarize images. Segmented images were then run through a custom MATLAB script to determine object density and the burden of the staining (% volume of 3D image stack occupied by the stain). Density values were then averaged to give an average burden. All custom software scripts are available on GitHub https://github.com/Spires-Jones-Lab. The MOAB-2 burden across all regions was averaged per slice to produce the final "slice burden".

## Plaque imaging in live HBSCs

HBSCs were imaged on a multiphoton microscope (Leica TCS SP8; Coherent Chameleon laser at 760 nm), using LAS X software, under a ×25 objective in a heat chamber (Oko Lab) set to 37 °C. Slices were imaged in Maintenance Medium to prevent changes in osmolarity or pH affecting live imaging readouts. Slices were pre-treated on top with either 1% Thioflavin-S diluted in Maintenance Medium or 100 μM Methoxy-X04 diluted in PBS for at least 20 min before imaging. Slices were imaged for a maximum of one hour, following which they were returned to the incubator for further culturing.

## Electrophysiology recordings

Whole-cell patch clamp recordings from 7 *div* HBSCs were made on a Scientifica-based recording rig, with recordings amplified using a MultiClamp 700B amplifier (Axon Instruments), and digitised using a Digidata 1550B using pClamp 11.2. Slices were removed from the culture membrane and placed into aCSF (33 °C) made of: 125 mM NaCl, 2.5 mM KCl, 25 mM NaHCO₃, 1.25 mM NaH₂PO₄, 25 mM Glucose, 1 mM MgCl₂, 2 mM CaCl₂, 1 mM Na-Pyruvate, 1 mM Na-Ascorbate. Following this, cells were targeted for whole-cell patch clamp recordings. Electrodes of 4–6 MΩ resistance were filled with a Potassium Gluconate based internal solution, made of in mM: 142 K-Gluconate, 4 KCl, 2 MgCl₂, 0.1 EGTA, 10 HEPES, 2 Na₂-ATP, 0.3 Na₂-GTP, 10 Na₂-Phosphocreatine. Biocytin was also included at a concentration of 0.1%, and the solution was adjusted to pH 7.35, at an osmolarity of between 290 and 310 mOsm. Cells were recorded gap-free in the current clamp for spontaneous synaptic currents and hyperpolarising and depolarising steps were elicited to evoke responses from the neuron.

## APOE genotyping

Samples were genotyped by the Edinburgh Genetics Core. All samples were genotyped using the TaqMan SNP Genotyping Assays. Taqman genotyping was carried out on the QuantStudio12KFlex to establish APOE variants using the following assays: C__3084793_10 for rs429358 and C___904973_10 for rs7412.

## BCA assay

Samples run on ELISA, Luminex® Multiplex Assay, or Western blot were assessed for protein content by BCA. For ELISA or Luminex® Multiplex Assay of culture medium, BCA analysis was also conducted on culture medium to normalise proteins of interest to the total secreted protein content present. For Western blot, or ELISA analysis of proteins present *within* the slice tissue itself, a BCA assay was conducted on the

slice tissue, with proteins of interest then normalised to the total protein content detected from the BCA.

A Piece MicroBCA Kit was used to assess protein content. In brief, bovine serum albumin (BSA) protein standards were provided for a range of 2 µg protein/µl to 40 µg/µl in duplicate, for a read volume of 200 µl. Test samples were then either diluted 1:200 or 1:400 (i.e. 1 µl in 200 µl or 0.5 µl in 200 µl). For culture medium samples, this allowed sufficient dilution of Phenol red to avoid interference with the colorimetric change. All standards and samples were then incubated at 37 °C for 1 hour and then read on a Clariostar plate reader at 562 nm according to manufacturer's instructions.

## ELISA

HBSC medium was collected after 7 *div*, flash frozen on dry ice in LoBind® Eppendorf's and stored at -70 °C until use. Total protein was measured using a micro BCA assay according to manufacturer instructions (ThermoFisher Scientific: 23235). ELISAs for $A\beta_{1-40}$ (ThermoFisher Scientific: KHB3481), $A\beta_{1-42}$ (ThermoFisher Scientific: KHB3544), total tau (ThermoFisher Scientific: KHB0042) and oligomeric Aβ (Biosensis, BSENBEK-2215-1P) were run according to manufacturers instructions. Culture medium was run undiluted for $A\beta_{1-40}$ and $A\beta_{1-42}$, whilst the medium was diluted 1/500 in standard dilution buffer (provided with ELISA kit) before tau analysis. Brain extract was run neatly for all Aβ forms, whereas it was serially diluted for tau. Values were obtained by comparing to a standard curve (pg/ml) then sample values were normalised to the total medium protein content (from BCA assay) to provide pg/mg. To assess the levels of Aβ in the slice tissue, slices were homogenised in 5 M guanidine hydrochloride (Sigma #G3272; 20 µl/slice) for 4 hours at room temperature. 180 µl/slice ice-cold PBS supplemented with 1x Halt Protease Inhibitor Cocktail (ThermoFisher Scientific: 78429) was then added to the guanidine extract, then spun for 20 minutes at 4 °C at 16,000×*g*. The supernatant was then run diluted (1/1–1/100) on the ELISA to accommodate differences in sample concentrations of Aβ. then read on a Clariostar plate reader. Values were obtained by comparing to a standard curve (pg/ml) then sample values were normalised to the total slice protein content (from BCA assay) to provide pg/mg.

## RNA isolation and quantitative reverse transcription PCR (RT-qPCR)

HBSCs (8 pooled slices) were removed from the culture membrane at 7 *div* with a scalpel blade into pre-filled bead mill tubes (Fisherbrand: 15555799) containing RNA*later* stabilisation solution (ThermoFisher: AM7020) and kept at 4 °C until RNA extraction. RNA was extracted using the TRIzol Plus PureLink RNA Purification Kit (Invitrogen: 12183555) which is a mixture of guanidinium thiocyanate-phenol-chloroform extraction and silica-cartridge purification methods. After removal of RNA*later*, samples were homogenised in 1 mL of TRIzol using a Bead Mill 24 Homogeniser (Fisherbrand: 15515799) to ensure equal homogenisation. Since HBSCs have high-fat content, the lysates were centrifuged for 5 mins at 13,000 rpm at 4 °C to pellet fatty tissue and cell debris, then the clear supernatant was transferred to a new tube for processing. RNA was quantified using a NanoDrop 2000 spectrophotometer (ThermoFisher: ND2000). To remove DNA contamination before qPCR, RNA samples were treated with Turbo DNAse (Invitrogen: AM2238) and the reaction was stopped with UltraPure EDTA (ThermoFisher: 15575020) at a final concentration of 15 mM. Quantitative RT-PCR was performed in 20 µl reactions in 96-well plates (ThermoFisher:AB0800W) using BRYT Green Dye contained in the GoTaq 1-Step RT-qPCR kit (Promega: A6020) in the CFX96 Touch Real-Time PCR Detection system (Bio-Rad). We used 100 ng of RNA per reaction and added extra $MgCl_2$ (ThermoFisher: R0971) at a final concentration of 3.25 mM (including the 2 mM $MgCl_2$ contained in the Master Mix), to each reaction to counteract EDTA chelation from the DNAse treatment (EDTA concentration in each

qPCR reaction was 3.75 mM). Primers (ThermoFisher) (detailed in Supplementary Table 4) were used at a final concentration of 250 nM. Ct values were normalised to the housekeeping gene GAPDH, which was amplified in parallel. Melt curves were checked to ensure the correct amplification of a single PCR product. The $2^{-\Delta\Delta CT}$ method was utilised to calculate relative gene expression levels.

## Luminex® multiplex assay

Analysis of several dementia-associated proteins including KLK-6, NCAM-1, Neurogranin, and TDP-43 was performed using a multiplex ProcartaPlex Human Neurodegeneration Panel 1 9-plex bead immunoassay (Cat#EXP090-15836-901). Standards and reagents were all prepared according to manufacturers' recommendations (Invitrogen ProcartaPlex manual). Culture media was diluted 1:2, 1:10, 1:50 and 1:100 in universal assay buffer and run in single. 25 µl of sample, standards or blanks was loaded onto the supplied 96-well plate along with 25 µl of universal assay buffer. The plate was incubated for 2 hours with antibody-coupled magnetic beads. The rest of the protocol followed standard ELISA conditions as per manufacturers' recommendations. The plate was analysed using a Luminex® 200 plate reader and data was generated using xPONENT software, and processed using the ProcartaPlex Analysis App on the Luminex® ThermoFisher Connect Platform. Dilution for each analyte was selected based on having sufficient reads on the standard curve for all samples with as little extrapolation as possible. The dilutions used were as follows: KLK-6 1 in 2, NCAM-1 1 in 50, Neurogranin 1 in 50 and TDP-43 1 in 2.

## Western blots

Acute tissue was collected or HBSCs were removed from the culture membrane with a scalpel blade, with both placed into RIPA buffer (ThermoFisher Scientific: 89901) with protease inhibitor cocktail (1X) and EDTA (1X) (ThermoFisher Scientific: 78429). Slices were thoroughly homogenised via pipette trituration. A BCA assay was used to assess protein concentration, and normalised stocks were made. Stocks were then mixed into equal volumes of 2X Laemmli buffer (Merck: S3401-10VL) and boiled for 10 minutes at 98 °C. Each sample was loaded into 4–12% NuPage Bis–Tris gels (Invitrogen: NP0336BOX) before proteins were separated by electrophoresis using MES SDS running buffer (Invitrogen: NP0002). Proteins were then transferred onto PVDF transfer membranes using an iBlot 2 or 3 machine (Invitrogen: IB24002). Following, a total protein stain (Li-Cor Biosciences: 926-11016) an image was acquired using a Li-Cor Odyssey Fc machine and then the membrane was de-stained. Membranes were subsequently blocked for 1 hour using PBS Intercept Blocking Buffer (Li-Cor Biosciences: 927-70001). Primary antibodies (see Supplementary Table 5 for details) were diluted in PBS Intercept Blocking Buffer with 0.1% Tween-20 and incubated with membranes overnight at room temperature, with shaking. Membranes were washed three times for 5 mins with PBS-Tween, then incubated in darkness for 2 hours with IRDye secondary antibodies (see Supplementary Table 5 for details), specific to the corresponding primary antibody species. Membranes were washed 3× in PBS supplemented with 0.1% Tween, 1× in PBS and then imaged using a Li-Cor Odyssey Fc machine. Western blot images were analysed using Empiria Studio (Version 2.3), with the software generating background-subtracted intensity signals. Proteins of interest were normalised to housekeeping proteins run on the same membrane.

## Lactate dehydrogenase (LDH) cytotoxicity assay

A subset of HBSCs (*n* = 3 slices from 3 cases) were treated with 1% Triton-X for ~8–10 hours to trigger maximal LDH release and were used as positive controls. The cyQUANT LDH cytotoxicity assay (Invitrogen; C20300) was used to quantify LDH released and was performed according to manufacturer instructions. Following the assay, a cytotoxicity % was produced using the following formula: % cytotoxicity = (sample LDH/ maximum LDH value on assay)×100.

## Statistics

All data was analysed using R and R Studio. Statistical tests were chosen according to the experimental design and dataset type.

The majority of statistical analyses used mixed effects models ('lme4' R package), as this allowed us to test if the treatment group impacted our variable of interest while controlling for pseudoreplication by including random effects. The dataset was assessed for normality, linearity and homogeneity of variance. If the data failed these criteria, then the data was either transformed using the method that was best for each individual model to fit the assumptions or a generalised mixed effect model was used and the family for the model was changed to fit the distribution of the data. Data transformations included square root, log, arcsine square root, and Tukey transformation and data families included Gamma, Poisson and Binomial. Post-hoc testing was conducted for pairwise comparisons, estimated marginal means and 95% confidence intervals ('emmeans' package), with Tukey correction for multiple comparisons. When data was transformed to meet the model assumptions then the reported effect sizes are computed on the back-transformed data. Statistical details for each individual analysis can be found in the results text. GLMM = Generalised linear mixed effects models, LMEM = linear mixed effects models, LM = linear model. Significance values were reported as $p < 0.05$*, $p < 0.01$*, $p < 0.001$***.

Correlation analyses were performed using the 'stats' package. Datasets were assessed for normality and if these criteria were met, a Pearson's correlation was used. If not, a Spearman's correlation was used. Significance values are reported as $p < 0.05$*, $p < 0.01$*, $p < 0.001$***. Chi-squared tests were performed using the stats package.

## Reporting summary

Further information on research design is available in the Nature Portfolio Reporting Summary linked to this article.

## Data availability

Source data are provided with this paper and uploaded to the University of Edinburgh DataShare Repository (https://doi.org/10.7488/ds/7890). Please note, to strengthen patient confidentiality, source data and Table 1 display the age of each patient to the nearest 5 years, whilst the statistical analysis in this study has been conducted on exact ages (to the nearest year). If exact ages are required, please get in touch with the corresponding author to discuss. Researchers seeking to obtain exact patient ages for this dataset would need to request Caldicott approval from NHS Lothian. The Caldicott Guardian would assess the request based on the principles of GDPR. Original Western blot images are included in the Supplementary Data File. Due to storage size limitations on the Edinburgh DataShare repository, raw microscopy images are available from the corresponding author upon request. Source data are provided with this paper.

## Code availability

Image analysis scripts are freely available on GitHub (https://github.com/Spires-Jones-Lab) and on the University of Edinburgh DataShare Repository (https://doi.org/10.7488/ds/7875).

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

## Acknowledgements

We thank the Lothian NRS (NHS Research Scotland) BioResource and Tissue Governance unit and EMERGE Research Nurse team, particularly Allan Macraild, Ikeoluwa Adekoya, Anuka Boldbaatar and Sarah Risbridger, for obtaining informed consent and accessing cortical tissue samples from NHS patients. We thank the Alzheimer Scotland Brain and Tissue Bank and the Massachusetts Alzheimer's Disease Research Centre (grant 1P30AG062421-01) for access to human post-mortem brain tissue. Thanks to the Genetics Core at the Edinburgh Clinical Research Facility, University of Edinburgh, for genotyping the human brain samples. We thank Dr. Henner Koch, Dr. Faye McLeod and Dr. Daniel Erskine for their valuable discussions on protocol development and optimisation of human brain slice cultures. Most importantly, we would like to thank patients and their families for providing tissue donations, without which this work would not be possible. This work was funded by grants awarded to Dr. Robert McGeachan from the Wellcome Trust, as part of the Edinburgh Clinical Academic Track for Veterinary Surgeons (225442/Z/22/Z), Dr. Claire Durrant from Race Against Dementia (ARUK-RADF-2019a-001), The James Dyson Foundation, and the Alzheimer's Society (581 (AS-PG-21-006), and by grants to Prof. Tara Spires-Jones from the UK Dementia Research Institute (UK DRI-4004) through UK DRI Ltd, principally funded by the Medical Research Council. The confocal microscope was generously funded by Alzheimer's Research UK (ARUK-EG2016A-6) and a Wellcome Trust Institutional Strategic Support Fund at the University of Edinburgh. Danilo Negro is a student in the Translational Neuroscience Ph.D. programme and is funded by Wellcome grant 228327/Z/23/Z.

## Author contributions

C.D., R.M. and S.M. designed the experiments with valuable input from P.B. and T.S.J. R.M., S.M., L.T., J.C., D.N., C.B., K.H., J.T., J.R., F.G., Y.Y.C., J.E., L.M., D.K., T.S.J. and C.D. performed experiments and collected data. R.M., S.M. and C.D. analysed/interpreted the data and performed statistical analysis with valuable input from T.S.J. P.B. and I.L. obtained surgical human tissue samples and C.D., R.M., S.M., L.T., S.B. and C.B. processed tissue for slice cultures. C.D., P.B. and S.B. developed human brain tissue collection and culturing pipelines. R.M., S.M. and C.D. wrote the manuscript, and all authors contributed to editing the manuscript and provided feedback. All authors approved the final version of this manuscript.

## Competing interests

Claire Durrant has no direct conflicts of interest with this study but receives funding from ONO Pharmaceuticals for a separate project. Tara Spires-Jones has no direct conflicts of interest with this study but has received payments for consulting, scientific talks, or collaborative

research over the past 10 years from AbbVie, Sanofi, Merck, Scottish Brain Sciences, Jay Therapeutics, Cognition Therapeutics, ONO Pharmaceuticals, and Eisai. She is also a Charity trustee for the British Neuroscience Association and the Guarantors of Brain and serves as a scientific advisor to several charities and non-profit institutions. The remaining authors declare no competing interests.
