## [Transparent Peer Review file · Nature Communications]

Divergent actions of physiological and pathological amyloid- β on synapses in live human brain slice cultures

Corresponding Author: Dr Claire Durrant

Version 0:

Reviewer comments:

Reviewer #1

(Remarks to the Author)

The authors report an interesting and important approach to address the translational gap between post mortem and non-human tissue-based studies of amyloid/tau physiology/pathophysiology and living human CNS tissue.

Over the last years several protocols have been developed enabling human CNS tissue derived from neurosurgical procedures to be maintained as slice cultures for extended periods of time. Such cultures open an opportunity to model disease pathogenesis in primary human CNS tissue and to overcome road blocks and limited translational success faced by conventional model systems.

Data reporting is sound and transparent with absolute values being provided in the Suppl. Material in addition to the normalized data of the main figures. Analysis of imaging, particularly array tomography imaging, was done in a blinded way which appears to be crucial.

A) In a first step the authors performed a characterization of the general viability of the slice cultures throughout a time span of 1 week. Structural maintenance of neurons, microglia and astroglia is demonstrated by immunohistochemistry. Functional integrity of neurons is exemplified by qualitative electrophysiology (demonstration of action potential firing and of synaptic transmission recorded in single neurons). Notably, microglia resemble a ramified non-activated (surveillant/resting state) morphology which is remarkable given the recent surgical procedure and tissue manipulation during slicing.

B) Next, the model was further characterized and basal release of A β 1-40, A β 1-42 and tau (stemming from the brain slice cultures) into the culturing medium was demonstrated. Relation of the respective levels to donor age/sex, brain region or APOE genotype over the culturing period of 7 days was analyzed and was shown to be largely independent from these factors, except for tau being significantly higher in tissue samples derived from temporal lobe and a tentative decrease of A β 1-40 with older age.

C) Very interestingly the authors demonstrate that A β release can be modulated (increased or decreased within the time window of the 1-week culturing process) by application of metalloprotease or beta-secretase inhibitors. Whereas APP levels and insoluble A β (measured by Western blot/ELISA) remained stable despite the pharmacological treatments the percentage area of A β within the slices (as assessed by immunohistochemistry) indeed paralleled the modulated A β release.

Further analysis found increased synaptic transcripts (such as synaptophysin) accompanying induced A β release, although this increase could not be pinpointed on protein level.

D) Next, cultures were treated with pathological (AD) brain extract. Brain extract immunodepleted from A β served as an important extra control additionally to straight medium treatment. Array tomography imaging indicated binding of pathological oligomeric A β to postsynaptic compartments and concomitant loss of presynapses.

E) In a last step, it was demonstrated that A β pathology is retained throughout the 1-week culturing period and A β plaque or tau pathology was shown to either tentatively or significantly go along with an increased A β 1-42/A β 1-40 ratio.

The authors draw the overall conclusions that pharmacologically raised physiological A β concentrations (which were found to enhance levels of synaptic transcripts) and exposure to pathological A β (shown to colocalize with postsynaptic domains

and being associated with loss of presynaptic structures) result in opposing effects on synapses in human brain tissue.

Ad A)

Immunohistochemical (Map2 and/or NeuN) and electrophysiological analysis of tissue from each case would be an important quality control and respective statements should be included. Currently it remains somewhat unclear whether viability of slices/neurons was only assessed during the initial experiments or whether these controls were part of the overall study design and performed throughout the course of the experiments.

Ad B)

A β 1-40 was found to be tentatively lower in samples derived from older patients. Were there any obvious age-related differences in synaptic density?

Ad C)

Induced A β release was associated with an increase of synaptic transcripts. It is possible that the enhanced levels of synaptic transcripts would escape a quantification on protein level (due to e.g. dilution effects by thousands of not affected cell types in the slices) and that a redistribution of protein between different cellular compartments would already be sufficient to achieve structural remodelling at the synapse. The ability to modulate A β release in human brain tissue is an important achievement and opens the possibility to study the downstream effects. So far, the authors remain on a descriptive level. Particularly, given that pathological A β (derived from patients with AD) was reported to result in a loss of presynapses it would be interesting to investigate whether an increase of physiological A β and concomitant enhanced levels of synaptic transcripts translate for example into increased synaptogenesis or protection from synaptic loss induced by pathological A β . I acknowledge that there is a chance that the 7-day time window of culturing in combination with very low synaptic turnover could prevent such an analysis. However, given that synaptic loss could indeed be monitored I wonder whether the authors explored this question?

Ad D)

I find it hard to judge from the array tomography data if the number of synapses per slice affected by oligomeric A β would be high enough to make a functional impact. Also, heterogeneity of the slices might prevent such an analysis but since the authors performed gap free recordings of synaptic input it seems intriguing to ask whether the reported synaptic loss could potentially be picked up on electrophysiological level? Can the authors please comment on this.

How did the authors deal with the quite prominent lipofuscin-related autofluorescence in human tissue?

Could upregulation of transcripts related to presynaptic compartments (following induction of A β release) be part of a repair process? This could be interesting in light of the described trend for an increased A β 1-42/A β 1-40 ratio (potentially due to a tentative decrease of A β 1-40 with older age and related lower repair capacity) in samples with plaques.

Reviewer #2

(Remarks to the Author)

The manuscript entitled "Opposing roles of physiological and pathological amyloid- β on synapses in live human brain slice cultures" by Robert McGeachan and colleagues reports that live human brain slice cultures (HBSCs) release Ab and tau in an age-, gender-, brain region- and Apoe genotype-dependent manner and postulate HBSC as a translational drug screening tool. Overall, this study provides some results to further establish HBSCs as a tool for translational purposes. However, the study appears to be premature and preliminary and there are several concerns that limit my enthusiasm for this study. While the method/tool is interesting, there are a number of outstanding questions that remain that would require significant experimental work to address in order to support their conclusion.

Main concerns:

- Although the subject is of interest, it has been already addressed in other studies with similar HBSC models, leading to similar conclusions. An example is provided by the work of Sebollela et al. in 2012 and Mendes et al. 2018, which addressed the effect of amyloid- β oligomers on adult human brain slices and described HBSC cultures already as a model for translational studies of neurodegenerative diseases. In my opinion the added value of this manuscript is minimal and I feel that the question regarding the novelty of this study seem to be more appropriate for a specialized readership.
- The model presented here has several limitations. They used in total 29 cases, however only 22 cases were screened for AD-associated pathology (Figure 5). It is not clear to me why not all 29 cases were included here? Even more problematic is the fact that out of these 22 screened cases, 18 % had indeed extracellular Ab plaques. I wonder if the missing 7 cases also had Ab plaques? Nevertheless, all cases with extracellular Ab plaques (and also tau tangles) should have been excluded from the main experiment and analyzed separately and treated as a distinct group.
- Along the same line how would the authors distinguish between plaque-derived Ab in the medium and exogenously applied oligomeric A that stems from human brain homogenate? Indeed, in line 342 the authors state that 1 case which had a high plaque burden in the tissue had also the highest Ab 42 levels in the medium. This is not a surprise to me and again indicates that all cases with Ab plaques definitively need to be excluded and treated as a separate group in order to obtain a 'purer' cohort for the main experiments.
- In addition, it is not enough to cite the literature for the oligomeric Ab composition of the applied brain homogenate,

because each human brain sample might differ in terms of Ab species. Therefore, each donor for brain homogenates needs to be analyzed and the species composition determined.

- One major concern is that the obtained data are over-interpreted and sometimes even misinterpreted. All raw-data displayed in Supplementary Figure 1 and 3 show no significant differences. Only when normalized, differences were significant in some instances. If one looks however closer, it becomes obvious that significance is often only reached due to 1 single case (e.g. Figure 4 c and d vs. raw data supplementary Figure 3).

- The Table with details about the cases on page 14 lists in most of the cases as reason for surgery glioblastoma. Therefore, it should be at least mentioned in the discussion that cases are not from healthy but diseased brains with glioblastoma which are known to display immunreactions that could in turn influence or have an impact on the results.

Reviewer #3

(Remarks to the Author)

In this manuscript, the authors used live human brain slice cultures (HBSC) from several patients undergoing surgery to remove primary or metastatic brain tumors. During this process, the surgeon resects non-malignant brain tissue to access the tumor. The authors use this access tissue to keep slices in culture and perform experiments to study processes related to Alzheimer's disease (AD), focusing on the release of amyloid beta peptides, tau, and the relationship of these with known risk factors for AD, namely APOE status and age. The paper could be divided into 1) the basal status of these variables in the culture system, and 2) testing how amenable the HBSC are to pharmacological interventions to study the effects of amyloid beta (Ab) in the brain, particularly on synapses, and potentially use them as a translational tool.

I would like to congratulate the authors for their work. The manuscript is well written, the findings are presented in a logical manner, the results are interesting and overall the quality of the paper is very good (I am particularly impressed by the statistics employed). The many challenges that pre-clinical models of dementia/AD face make this study timely and of critical importance. I think such models could be very informative when testing the mechanisms behind novel therapies.

I have a few comments aiming at, in my opinion, increasing the clarity of the manuscript for the reader. They are provided in the same order that the data is presented in the manuscript.

1. In the introduction, the authors contextualize their work and the importance of translational tools to study AD such as HBSC by citing work done in human samples (cerebrospinal fluid, interstitial fluid, brain imaging) and in in vitro models (mostly rodent AD transgenic models and induced-pluripotent stem cell derived neurons). Previous work done in human brain slices goes uncited at this point, despite of many years of research and optimisation by other groups. A more balanced introduction might help the reader in contextualizing the present findings. Work cited in the discussion could be cited here.

2. In Fig.1, the authors show images of several markers in the cultures at 7 DIV to show the preservation of cellular diversity and function in the HBSCs. They showed the presence of neurons, astrocytes and microglia. I wondered if:

a. Did the authors do stainings of slices at different DIV to show how the presence of these cell types might change, or are maintained, during the culture process? For example, markers of different neuronal types (e.g Satb2 for excitatory neurons, VGlut1 or NMDAR1 for glutamatergic neurons). These data would be interesting to understand how neuronal populations with different levels of vulnerability to AD-pathology remain in the slices, opening up the exciting possibility to study the responses to interventions of specific populations.

b. Did the authors monitor cell viability over the course of the experiment with, for example, an LDH assay? Knowing the health of the culture at the time of measurement of the release of tau and other proteins would be important for the interpretation.

c. Particularly relevant for glia, did the authors use any marker of activated microglia (such as CLEC7A) over culture? Comparing the levels of such markers at least at 0 DIV (acute slice) with 7 DIV could give some insight into the behavior of cell types during the culture process (particularly, the potential inflammation induced by slicing the tissue). By studying inflammation-related markers in glia cells and cytokines released into the media, one would know whether the culture undergoes acute inflammation that decreases over time, and dictate what would be the best DIV to start conducting experiments without that potentially cofounding factor?

d. Fig.1g: I found it difficult to see the astrocytic endfeet at this magnification and without a marker of endothelial cells in the vessel. Providing a higher magnification image would be helpful.

e. Fig 1h,i: Since electrophysiology is not within my areas of expertise, I found hard to track the data herein presented: Are the synaptic activity and action potentials showed from a representative neuron? How many slices and donors were tested? How many cells? I could not find any of this in the figure legend nor the material and methods. A more integrated data reporting within the manuscript would aid in data interpretation for the reader (and perhaps be helpful for addressing comment 6,d below).

3. Fig2b,c: There is a remarkable outlier in Ab42 levels (a female case). The authors point in the discussion to a case with very high levels of Ab42, which I think might be this one, having also the higher burden of plaque or amyloid pathology. As the authors discuss, it is indeed intriguing that this case would have the highest levels of soluble Ab42, opposite to what one would expect based on other models/human in vivo samples. Do the authors have any data to investigate if this might due to

biological or technical variability? For example, is there information on the cases' clinical data that might help in identifying possible reasons (such as epilepsy, cognitive assessments...)? Could they rule out possible technicalities affecting the Ab42 levels of that sample (infection, inflammation...)?

a. Fig.2i: It seems as the females in the APOE3/3 group have lower levels of tau released into the media compared to males? As Age was one of the variables (random effects) included in the model, if I understand correctly, sex-dependent differences would have been found?

4. Fig.3 shows the effects of pharmacologically modifying amyloid beta levels:

a. Did the authors test a solvent-only control alongside the medium-only control? Avoiding this control might skew the results as is common to have effects by organic solvents (in this case, DMSO for BACE1 inhibitor).

b. The concentration of phosphoramidon is very high - was this required due to penetration issues into the slice? Did the authors try several doses to find the one that would elicit the best response and minimise off-target effects?

c. Whilst the data clearly shows a decrease in pre-synaptic density when soluble amyloid derived from post-mortem AD brain was applied (Fig.4), supporting the damage and loss of pre-synaptic compartments previously reported in AD, the data to support the opposite (and that sustains the title of the paper) is limited to the upregulation of SYP mRNA. Notably, the authors did not find an upregulation at the protein level (Suppl. Fig.2). The treatment was applied for 7 days (is this correct?), a timeframe during which one might expect to see an effect at the protein level? The discussion of these data in lines 370-379 could be a place to discuss these observations. Adding a point in the discussion about future plans to understand the impact on pre-synaptic compartments I think would be a nice addition.

d. Further to comment 4c, Sebollela et al (PMID: 22235132) reported a decrease in synaptophysin transcripts in adult human cortical slices exposed to sublethal doses of amyloid beta oligomers (A β Os, 500nM). I am curious about this data in relation to the authors finding in their phosphoramidon-treated slices of an increase in synaptophysin (SYP) mRNA (Fig.3g). These findings are discussed in lines 386-388 but not in relation to the phosphoramidon data. Do the authors think the opposite effects are due to a lower amount of amyloid beta reached by this treatment compared to the concentration of synthetic amyloid oligomers used in the other study? The MOAB-2 antibody could recognise soluble oligomers, and they do detect and increase in MOAB-2 positive area, so I would expect and increase in soluble oligomers with phosphoramidon treatment. Since the authors measured the levels of amyloid peptides in the culture media in these experiments (presented in Supp Fig3 as normalised datapoints), I wondered if they could include the expected final amounts of amyloid peptides reached that elicited the found effects on synaptic transcripts to facilitate comparison to other studies. To specifically find out the final concentration of oligomeric amyloid, they could use ELISA kits with the MOAB-2 antibody as capture and detection if there was an opportunity to do so (<https://uk.vwr.com/store/product/17799539/oligomeric-amyloid-a-elisa-kit>).

5. Fig.4 shows the effects of amyloid beta enriched or depleted extracts from AD brains on synapses using array tomography. They find that whilst amyloid co-localised more often with post-synaptic terminals, the number/area covered by those (PSD95) did not change. Instead, the number of pre-synaptic terminals was reduced. This is an important and interesting finding. Beyond the basics, such as finding it an appropriate imaging technique to study synapses, a detailed critical revision of the array tomography data herein presented lies beyond my expertise. Only two comments:

a. Do the authors have any characterization of the AD-soluble extract preparation to show, such as western blot? Even if supplementary, there is a lack of characterization of the material used, and I think is important to discern the impacts of soluble (monomeric, oligomeric...) and insoluble amyloid on synapses.

b. It's intriguing that despite a decrease in pre-synaptic terminal area, there was no change in the synaptophysin levels in the lysate. Did the authors blot for other pre-synaptic proteins? Perhaps western blot of the whole homogenate might not be sensitive enough. Alternatively, the authors suggest this might be due to a re-distribution of the pool of synaptophysin protein rather than affecting its levels. If there was material available, the authors could do a fractionation experiment to isolate a synaptosomal-enriched fraction and check whether the levels of such fraction are decreased in Ab+ compared to Ab-, and if the levels of synaptic proteins are disproportionately affected.

6. In Fig.5 and Fig.6, the authors show data regarding the endogenous and spontaneous presence of pathological changes linked to AD in the brain slices, namely the aggregation of amyloid species and the phosphorylation of tau at AD-specific sites. The authors show evidence of their presence in the slices and their maintenance after 7DIV, which is appreciated. As expected, those increased with age. Intriguingly and interestingly, they report an increase in Ab42:40 in those slices with higher plaque burden. This is contrary to what one might expect based on the literature in CSF and even ISF. I agree with the discussion (lines 350-352) about analyzing fluid samples and slices derived from the same patient in order to understand this phenomenon better in future work. A few comments:

a. Since none of the cases had been diagnosed with a neurodegenerative disease, and given the relatively young age of the cases, would it not be difficult to discern between early pre-symptomatic dementia, or the natural appearance of amyloid and tau in aging individuals (such as reported by Elobeid et al, 2016)? Compared to studies in NPH patients, which have a higher incidence of dementia (one in two cases) than the expected in the rest of the population, the cases used in this study are not expected to be enriched in early, pre-symptomatic AD cases.

b. Fig.5b,c: Did I understand correctly that there was no case in which there was co-occurring fibrillary Ab and phospho-tau

detected? Since none of the cases had been diagnosed with AD, one might not expect it, but wanted to make sure. To aid the reader, a table summarizing the cases and their characteristics (age, sex, others) related to the amyloid- and tau-pathology status showed (and any potential overlap) would be helpful.

c. Since effects on synapses might be expected early in the pre-clinical phase of AD, I am curious to know if the authors performed any synaptic staining in these slices to see if there is any early effect? I also wondered if they have looked for the presence of dystrophic neurites.

d. Since the authors studied the electrophysiological characteristics of the slices in Fig.1, I wondered if they could mine that dataset for the slices with and without amyloid/tau pathology and see if there are early signs of decreased synaptic activity?

e. The authors discuss the possibility that their ELISA kits are not sensitive enough to detect decreases in Ab42 with age (line 350). Would an MSD assay be more suited to study these samples? I think it has lower LLOD and might be more sensitive to small changes.

f. If there was media left from these experiments, I would be curious to understand how HBSC recapitulate other early pre-clinical AD processes. Would be great to study the levels of other fluid biomarkers in the media (e.g. NfL, synaptic health markers such as neurogranin...) and their relationship with DIV, age of the donors and Ab levels.

Version 1:

Reviewer comments:

Reviewer #1

(Remarks to the Author)

The authors took substantial effort to address my comments. Several new data sets have been acquired and added to the manuscript (array tomography characterisation of human brain slice response to BACE1 inhibitor/ phosphoramidon treatment, qPCR analysis of transcript changes in response to pathological A β -containing AD brain extract, analysis of additional proteins in culture medium, examination of sex effects, additional analysis of relationships between neuronal/synaptic protein and A β / tau release), which strengthen the paper.

In the first version it remained somewhat unclear whether viability of slices/neurons was only assessed during the initial experiments or whether these controls were part of the overall study design and performed throughout the course of the experiments.

The authors now clarified this point and added additional quality control data assessing macroscopic appearance of the tissue, LDH release over time as a marker of cytotoxicity and expression of synaptic and neuronal proteins during culturing.

"A β 1-40 was found to be tentatively lower in samples derived from older patients. Were there any obvious age-related differences in synaptic density?"

The authors now explored this question by quantifying the levels of synaptic markers synaptophysin and PSD95, and neuronal proteins Tuj1 and PGP9.5 using Western blot in correlation with donor age, excluding an age-dependent alteration of synaptic markers (furthermore, whereas Tuj1 was found decreased, PGP9.5 was not). Furthermore, there was no correlation between synaptic and neuronal protein levels with A β 1-40, A β 1-42 or tau in the culture medium.

"Particularly, given that pathological A β (derived from patients with AD) was reported to result in a loss of presynapses it would be interesting to investigate whether an increase of physiological A β and concomitant enhanced levels of synaptic transcripts translate for example into increased synaptogenesis or protection from synaptic loss induced by pathological A β ." The authors now added new experiments addressing these questions. Interestingly, the set of new data showed a loss of synaptophysin puncta following manipulation of physiological A β levels in either direction. An analysis of effects of pathological A β -containing AD-brain extract on synaptic transcripts, furthermore, did not show any change in synaptic transcripts. These data highlight the complexity of the biological processes. The discussion has been expanded accordingly.

"I find it hard to judge from the array tomography data if the number of synapses per slice affected by oligomeric A β would be high enough to make a functional impact."

Additional electrophysiological experiments could not be added due to limited access to human tissue samples and AD-brain extract. The authors now discuss such experiments as possible directions of follow-up studies.

"How did the authors deal with the quite prominent lipofuscin-related autofluorescence in human tissue?"

The respective information has been added to the methods section.

"Could upregulation of transcripts related to presynaptic compartments (following induction of A β release) be part of a repair process?"

The discussion has been expanded to cover these points, particularly in light of the newly added data on modulation of physiological levels of A β and the respective effects on synapses.

Reviewer #2

(Remarks to the Author)

The response to reviewers and the revised manuscript satisfactorily addresses my comments and criticisms raised during the review process. I have no additional concerns.

Reviewer #3

(Remarks to the Author)

I would like to first thank the authors for their efforts in addressing the reviewer's comments during the first round of revision. The authors have largely addressed the comments I provided, and I am very pleased that addressing these has strengthened the paper and their conclusions, particularly on the potential hermetic role of Abeta in synapses.

I will first summarise the points that were successfully addressed and end with minor points that, in my opinion, would still require some amendments or clarification for the reader.

By addressing the concerns raised in response points 16 and 27, I think the new version of the manuscript better contextualises the findings with the relevant previous literature.

In response to point 18, the authors have now included LDH assay data to support the viability of the cultures. Importantly, they also added LDH data upon treatment with BACE1 inhibitor and phosphoramidon, which I think is an important addition and addresses my comments in points 24 and 25. The authors discuss that due to the limited availability of the tissue, they do not routinely assess viability with this assay to avoid sacrificing valuable slices for the 1% Triton X-100 positive control. I appreciate how the most accurate way to assess maximal toxicity is to have a positive control for each case. However, now that the authors have done LDH assays for several cases, I wonder if they have observed such a big variability between positive controls from different cases? If not, they could consider making a batch from multiple cases and re-use this positive control for all their LDH assays. I would prioritise this type of analysis because it requires little amount of volume and, due to their interest in physiologically secreted proteins, membrane permeability would be an important cofounder to take into account.

Response points 19, 20, 22, 23, 30, 31, 32, 34: thank you for your efforts in addressing these minor points (including my poorly worded question in point 23).

Response point 26, also raised by reviewer 1: I agree with the authors that the new data on the modulation of amyloid beta and its effects in synapses in living human brain tissue is of importance and a great starting point to study the role of amyloid in synapses in a translational model. I appreciate that the authors tried measuring the levels of oligomeric amyloid in the samples in point 27, and I appreciate how the low levels in the sample would be a technical challenge. I also think is positive that they added new discussion points to address the differences with previous studies.

Response point 28: Thank you for the characterisation data of the AD-soluble extract. I think this is an important addition to present and future experiments using AD-soluble extract to treat slice cultures to model AD-like processes.

Minor points I think would need addressing:

In response to points 17 and 21, the authors have included a small characterisation of how neuronal (focusing on synaptic) proteins change over time in culture (acute vs 7DIV) in two cases of the cohort, adding new data into Supp Fig 1 and manuscript lines 136-140.

- The western blots in the red channel at the bottom of Supp. Fig.1c are unlabelled, could the authors please label these?

- Quantifications show significant decreases in neuronal proteins that are commented in the main text as "modest". I would disagree with this, particularly for PSD-95, and just conclude that for some proteins, the resection and culture conditions had a significant impact.

- Do the authors think that the impact on synaptophysin and PSD-95, seemingly greater for the later, may have had any impact on the results of the paper?

Could the authors please clarify which cases were used for Fig.1 – they are mentioned as a subset of cases, so I understand must be within the 42 cases in Table 1; however, no case has been allocated to Fig. 1 in this table. Are perhaps the cases attributed to Supp. Fig.1 the ones also used in Fig. 1? (n=11, in line with the numbers mentioned in the main text). In the rebuttal letter these cases are referred to as "characterisation pilot", which made me think they might be independent of the main cohort?

As a minor point, I have noticed some proteins are named inconsistently throughout the manuscript (e.g, PSD-95 and PSD95; TDP-43 and TDP43).

Response point 29: Interesting that synaptophysin and synapsin-1 show different effects, I wondered what are the authors' thoughts on this finding? I appreciate experimentally addressing this is beyond the scope of the current manuscript.

Response point 35: I am pleased that the revision of the manuscript provided further insights into how brain slice cultures could be used to study the release of fluid biomarkers of neurodegeneration.

I noticed that the authors normalise the levels of the secreted proteins analysed by ELISA or Luminex to the total protein content in the medium and wanted to ask the authors about this. In our research, colleagues and I have also looked for an appropriate way to normalise protein release measured by ELISA to account for the variability in the number of cells between conditions. Measuring protein content in media and lysates in parallel by BCA assay in our hands usually results in high absorbance values in media that do not seem to correlate to total protein differences detected in lysates – so as a proxy

for cell confluency, we usually normalise ELISA measures of secreted proteins to the total protein content in the lysates. Others measure genomic DNA in the samples and calculate an approximate number of cells in the culture counting that each cell would have approx 6 pg of DNA (<https://www.qiagen.com/us/resources/faq?id=1bfcd402-f33a-4d01-9090-83f150b47276&lang=en>).

The caveat with total protein content in media measured by BCA is that such measurements might be challenging due to the supplements present in media (which might contain proteins at higher concentrations than the secreted proteins by the culture, thereby reducing the sensitivity to the culture-derived proteins), and/or to the colour of the media (as in our case, there is phenol red present). Were the authors concerned by the same issues, and if so, how did they address them? Did they do dilutions of the media to address potential problems with phenol red? A more detailed description in material and methods would aid reproducibility of their techniques.

McGeachan & Meftah et al. Response to Reviewers

We would like to thank the reviewers for their time and providing constructive comments on our paper. We have added new experimental data, including additional Array Tomography characterisation of human brain slice response to BACE1 inhibitor/ Phosphoramidon treatment, qPCR analysis of transcript changes in response to pathological A β -containing AD brain extract, analysis of additional proteins in culture medium, examination of sex effects, and additional analysis of relationships between neuronal/synaptic protein and A β / tau release. Alongside text changes and methods clarification, we believe these changes have significantly strengthened the paper. We respond to point by point comments below.

Reviewer 1

- 1) **R1 summary of work:** “The authors report an interesting and important approach to address the translational gap between post mortem and non-human tissue-based studies of amyloid/tau physiology/pathophysiology and living human CNS tissue. Over the last years several protocols have been developed enabling human CNS tissue derived from neurosurgical procedures to be maintained as slice cultures for extended periods of time. Such cultures open an opportunity to model disease pathogenesis in primary human CNS tissue and to overcome road blocks and limited translational success faced by conventional model systems. Data reporting is sound and transparent with absolute values being provided in the Suppl. Material in addition to the normalized data of the main figures. Analysis of imaging, particularly array tomography imaging, was done in a blinded way which appears to be crucial.

A) In a first step the authors performed a characterization of the general viability of the slice cultures throughout a time span of 1 week. Structural maintenance of neurons, microglia and astroglia is demonstrated by immunohistochemistry. Functional integrity of neurons is exemplified by qualitative electrophysiology (demonstration of action potential firing and of synaptic transmission recorded in single neurons). Notably, microglia resemble a ramified non-activated (surveillant/resting state) morphology which is remarkable given the recent surgical procedure and tissue manipulation during slicing.

B) Next, the model was further characterized and basal release of A β 1-40, A β 1-42 and tau (stemming from the brain slice cultures) into the culturing medium was demonstrated. Relation of the respective levels to donor age/sex, brain region or APOE genotype over the culturing period of 7 days was analyzed and was shown to be largely independent from these factors, except for tau being significantly higher in tissue samples derived from temporal lobe and a tentative decrease of A β 1-40 with older age.

C) Very interestingly the authors demonstrate that A β release can be modulated (increased or decreased within the time window of the 1-week culturing process) by application of metalloprotease or beta-secretase inhibitors. Whereas APP levels and insoluble A β (measured by Western blot/ELISA) remained stable despite the pharmacological treatments the percentage area of A β within the slices (as assessed by immunohistochemistry) indeed paralleled the modulated A β release.

Further analysis found increased synaptic transcripts (such as synaptophysin) accompanying induced A β release, although this increase could not be pinpointed on protein level.

D) Next, cultures were treated with pathological (AD) brain extract. Brain extract immunodepleted from A β served as an important extra control additionally to straight medium treatment. Array tomography imaging indicated binding of pathological oligomeric

A β to postsynaptic compartments and concomitant loss of presynapses. E) In a last step, it was demonstrated that A β pathology is retained throughout the 1-week culturing period and A β plaque or tau pathology was shown to either tentatively or significantly go along with an increased A β ₁₋₄₂/A β ₁₋₄₀ ratio. The authors draw the overall conclusions that pharmacologically raised physiological A β concentrations (which were found to enhance levels of synaptic transcripts) and exposure to pathological A β (shown to colocalize with postsynaptic domains and being associated with loss of presynaptic structures) result in opposing effects on synapses in human brain tissue.”

Thank you for your kind and thorough summary of our work.

- 2) R1: “Immunohistochemical (Map2 and/or NeuN) and electrophysiological analysis of tissue from each case would be an important quality control and respective statements should be included. Currently it remains somewhat unclear whether viability of slices/neurons was only assessed during the initial experiments or whether these controls were part of the overall study design and performed throughout the course of the experiments. “

We have now clarified in **lines 126-128, 136 & 140** that immunohistochemical staining and 7 *div* electrophysiology was only performed on a subset of cases during the initial optimisation period as we established our in-house protocol. Once we were confident that we were obtaining consistent survival of cultures we no longer performed these assessments on every case. Quantity of tissue available from each case can be extremely limited, sometimes as few as a couple of dishes of slices, meaning experiments were prioritised over repeating previously conducted stains. For all experimental treatments, we used a repeated measures design, so all treatment effects are normalised to a non-treated control from the same tissue sample-meaning variability between cases is controlled for as much as possible. To further provide information on our quality control procedures, we have added a figure explaining our tissue assessment criteria, demonstration of LDH release over time in culture, and provided additional comparisons between acute tissue and 7 *div* cultures. (**Supp. Fig. 1**).

- 3) R1: “A β ₁₋₄₀ was found to be tentatively lower in samples derived from older patients. Were there any obvious age-related differences in synaptic density?”

This is an interesting point, and we have now provided additional data examining this. In acute tissue samples (prior to culturing), we assessed the levels of synaptic markers synaptophysin and PSD95, and neuronal proteins Tuj1 and PGP9.5 using Western blot and performed correlation analysis with age (**Supp. Fig. 2a**). We found a significant decrease in Tuj1 with age, but no significant changes with other markers. To further explore how synaptic and neuronal proteins may influence release of A β and tau, we examined the correlation between synaptic and neuronal protein levels in 7 *div* HBSC tissue and compared with the corresponding culture medium proteins. We performed this in the subset of our cases where we had both 7 *div* Western blot samples and culture medium samples available from the same case. There were no correlations between the levels of synaptophysin, PSD95, Tuj1 or PGP9.5 with the levels of A β ₁₋₄₀, A β ₁₋₄₂ or tau in the culture medium. This data is now displayed in **Supp. Fig. 2b-d**.

- 4) R1: “Induced A β release was associated with an increase of synaptic transcripts. It is possible that the enhanced levels of synaptic transcripts would escape a quantification on protein level (due to e.g. dilution effects by thousands of not affected cell types in the slices) and that a redistribution of protein between different cellular compartments would already be sufficient to achieve structural remodelling at the synapse. The ability to modulate A β

release in human brain tissue is an important achievement and opens the possibility to study the downstream effects. So far, the authors remain on a descriptive level. Particularly, given that pathological A β (derived from patients with AD) was reported to result in a loss of presynapses it would be interesting to investigate whether an increase of physiological A β and concomitant enhanced levels of synaptic transcripts translate for example into increased synaptogenesis or protection from synaptic loss induced by pathological A β . I acknowledge that there is a chance that the 7-day time window of culturing in combination with very low synaptic turnover could prevent such an analysis. However, given that synaptic loss could indeed be monitored I wonder whether the authors explored this question?"

Thank you for your acknowledgement of the importance of being able to modulate A β in a human system and your interesting experimental question. We have performed new experiments to assess the impact of physiological A β modulation on synaptic puncta using Array Tomography. This data is now displayed in **Fig. 4h-i**. Here we found a significant treatment effect on synaptophysin (but not PSD95) density following manipulation of physiological A β levels in *either* direction, with a significant loss of synaptophysin puncta detected. We also performed additional experiments assessing the impact of pathological A β -containing AD-brain extract on synaptic transcripts, which we now display in **Fig. 5g**. Interestingly, in contrast to the increase seen with Phosphoramidon treatment, we found no change in synaptic transcripts in response to pathological A β application. Taken together, these new data further highlight a potential hormetic role of A β , as well as divergent responses at the synapse level to small (picomolar) increases in physiological versus pathological A β . We provide expanded discussion of these findings in results **lines 253-256, 258-259, 312-314** and discussion **lines 476-480, 485-493, 495, 503-505**, including addition of the possibility of reorganisation/ compensation suggested by this reviewer.

- 5) R1: "I find it hard to judge from the array tomography data if the number of synapses per slice affected by oligomeric A β would be high enough to make a functional impact. Also, heterogeneity of the slices might prevent such an analysis but since the authors performed gap free recordings of synaptic input it seems intriguing to ask whether the reported synaptic loss could potentially be picked up on electrophysiological level? Can the authors please comment on this."

Although the percentage of synapses affected by application of pathogenic A β is low, given the sheer number of synapses within the brain, our effect size equates to a loss of around a mean loss of 102,528,403 synapses per mm³. The loss of these puncta is likely sufficient to impact neuronal network function, a point which we have raised in discussion **line 506-507**. Performing additional electrophysiology experiments in the timeframe for review was unfortunately unfeasible, due to limited additional human cases, a high number of cells required per case (meaning a large number of slices) to obtain meaningful recording data, and a limited supply of AD-brain extract from the stock we performed previous experiments in. We have added in a sentence in the discussion to highlight this point as a future avenue of research, as this would be a very interesting direction to explore as an independent study (**line 508-511**).

- 6) R1: "How did the authors deal with the quite prominent lipofuscin-related autofluorescence in human tissue?"

For immunohistochemical staining and analysis, an autofluorescence eliminator reagent was used as part of the staining protocol which we had optimised to help reduce autofluorescence

(Methods **line 810-811**). For Array Tomography staining and analysis, ultrathin sectioning means that true immunostaining is significantly brighter than lipofuscin autofluorescence, so easily segmented out during thresholding. Further, for immunostaining of synaptic proteins, we use size exclusion criteria during image segmentation (Methods **line 706-802**). Our repeated measures design for treatment experiments also means that autofluorescence is unlikely to impact treatment interpretations, as the baseline levels within slices from the same case are very similar.

- 7) **R1: “Could upregulation of transcripts related to presynaptic compartments (following induction of A β release) be part of a repair process? This could be interesting in light of the described trend for an increased A β 1-42/A β 1-40 ratio (potentially due to a tentative decrease of A β 1-40 with older age and related lower repair capacity) in samples with plaques.”**

This is an excellent point, and we, and others, agree that endogenous A β may have many physiological roles including involvement in synaptic reorganisation and repair. This also aligns with our new findings that increasing physiological (Phosphoramidon treatment to increase endogenous levels) or pathological (application of A β -containing AD extract) A β concentration leads to reduced synaptophysin puncta, but compensation through increased synaptic mRNA transcripts is only seen in response to physiological elevations in A β (**Fig. 4**). We also believe that our findings further support a hormetic role of A β , with deviations in concentration in either direction affecting synapses (as shown by loss of synaptophysin puncta in response to bi-directional endogenous A β manipulation). Importantly these changes occur without changes to LDH release, indicating cellular toxicity is not responsible for the observed puncta loss (**Supp. Fig. 4i**). We expand our discussion of these points in **lines 476-480, 485-493, 495, 503-505**.

Reviewer 2

R2 Summary of work: “The manuscript entitled “Opposing roles of physiological and pathological amyloid-b on synapses in live human brain slice cultures” by Robert McGeachan and colleagues reports that live human brain slice cultures (HBSCs) release Ab and tau in an age-, gender-, brain region- and Apoe genotype-dependent manner and postulate HBSC as a translational drug screening tool. Overall, this study provides some results to further establish HBSCs as a tool for translational purposes. However, the study appears to be premature and preliminary and there are several concerns that limit my enthusiasm for this study. While the method/tool is interesting, there are a number of outstanding questions that remain that would require significant experimental work to address in order to support their conclusion.”

Main concerns:

- 8) **R2: “Although the subject is of interest, it has been already addressed in other studies with similar HBSC models, leading to similar conclusions. An example is provided by the work of Sebollela et al. in 2012 and Mendes et al. 2018, which addressed the effect of amyloid-b oligomers on adult human brain slices and described HBSC cultures already as a model for translational studies of neurodegenerative diseases. In my opinion the added value of this manuscript is minimal and I feel that the question regarding the novelty of this study seem to be more appropriate for a specialized readership.”**

Thank you for highlighting these studies, we had previously referenced them as part of our discussion (original manuscript line 386-387). We agree that these are important studies, but

respectfully believe that the work described in our paper represents a significant step forward, and explores different questions which are of interest to a wide audience. We have now moved discussion of this paper, and others, to the introduction to earlier highlight previous foundational work in this area (**lines 105-110**), we also provide an extended discussion on these works in the discussion, comparing our findings directly with theirs (**lines 501-511**).

For the purpose of response to review, we highlight key areas of difference of our work. Compared to previous studies, our work is the first of its kind to: assess **endogenous** production of A β , tau and other neurodegenerative markers in the culture medium, and seek to explore the relationship with age, brain region, sex, *APOE* status, spontaneous pathology and protein levels in the tissue. We also provide the first demonstration that endogenous production of A β can be manipulated pharmacologically in human slices, allowing us to explore the consequences of manipulating physiological levels of A β . Our work applying exogenous A β (perhaps the closest work to that shown in Sebolella 2012/ Mendes 2018) is also unique in that we apply pM levels of A β derived from genuine AD cases, as opposed to high nM (250-500nM) concentrations of synthetic A β peptide. Our results differ in that we directly show A β binding to postsynaptic densities, and show a loss of pre-synaptic puncta through Array Tomography. In contrast to the Sebolella paper, we also do not see changes to synaptophysin mRNA (new data shown in **Fig. 5g**), highlighting that synthetic A β may result in different responses to those seen with experimental treatments more closely representing disease states. In addition to exploring differences between response to physiological and pathological A β , we also observe endogenous pathology in a subset of our samples- and link this to changes in the culture medium ratio of A β_{1-42} /A β_{1-40} . We have also expanded this study through revision experiments to examine other putative neurodegenerative biomarkers and how these relate to biological characteristics and pathology as well (**Fig 3 & 7**). Taken together we believe our work represents a major step-change in using human brain slice culture models to explore key questions relating to Alzheimer's disease and other associated disorders which will be of interest to a broad audience.

9) R2: "The model presented here has several limitations. They used in total 29 cases, however only 22 cases were screened for AD-associated pathology (Figure 5). It is not clear to me why not all 29 cases were included here?"

We have now screened all cases used in this paper for pathology (including those used for revision experiments) and have updated the tables and figures to reflect this (**Fig 6 & 7, Table 1**). The data will also be made available following publication and can be further pursued by interested researchers.

10) R2: "Even more problematic is the fact that out of these 22 screened cases, 18 % had indeed extracellular Ab plaques. I wonder if the missing 7 cases also had Ab plaques? Nevertheless, all cases with extracellular Ab plaques (and also tau tangles) should have been excluded from the main experiment and analyzed separately and treated as a distinct group."

We would like to refute the point that some of the cases having endogenous pathology is problematic. None of the patients included in this study have been formally diagnosed with a neurodegenerative disease and, aside from having to undergo tumour surgery, are a randomly selected group from the general population. It is likely many of the cases we see pathology in may one day convert to Alzheimer's disease (1/3 individuals will develop AD in their lifetime). Unfortunately we cannot know for certain if the cases in our study would convert to AD as the low life expectancy of individuals with brain cancers means we are unlikely to be able to

conduct long-term follow up. To effectively observe and model changes relevant to the onset of dementia, we need to use models that recapitulate the genuine human condition. In an aging human study we don't believe there would be such thing as a 'pure cohort' as most individuals will show some sort of pathological features during ageing. We have not screened for other pathology such as α -synuclein, TDP-43, vascular changes etc. so to exclude samples on the basis of tangles and plaques (without a diagnosis of AD) would be arbitrary and reduce validity when trying to capture changes that happen over the human lifespan. Such features are an important part of the unique patient profile, especially in the context of aging where this pathology is commonplace in healthy individuals over the age of 50 (See DOI: 10.1212/WNL.0b013e318245d295 and DOI: <https://doi.org/10.1007/s00259-021-05230-5>). We could separate our samples based on age, as pathology is only visible in samples from individuals over 50 in our cohort, but this would go against the aims of our study.

In this study, **we do separately assess** the impact of the presence of $A\beta$ and tau pathological features (**Fig 6&7**) and find very few changes. Here we report only minor changes to $A\beta_{1-42:1-40}$ ratio and KLK-6 release in response to tau tangles, and a trend for change in $A\beta_{1-42:1-40}$ ratio in individuals with $A\beta$ plaques. In treatment experiments, all effects are normalised to non-treated control slices from the same individual, so the impact of any between-case differences is controlled for. For transparency, we have now updated our case table to include details of pathology and what experiments they were used in (**Table 1**). The majority of cases within the treatment experiments did not display pathological features.

11) R2: "Along the same line how would the authors distinguish between plaque-derived Ab in the medium and exogenously applied oligomeric $A\beta$ that stems from human brain homogenate? Indeed, in line 342 the authors state that 1 case which had a high plaque burden in the tissue had also the highest Ab 42 levels in the medium. This is not a surprise to me and again indicates that all cases with Ab plaques definitively need to be excluded and treated as a separate group in order to obtain a 'purer' cohort for the main experiments."

In this experiment (data displayed in **Fig. 5**) a repeated measures design was used. Here, slices from the same case were used in the medium only control, $A\beta$ negative and $A\beta$ positive group. This accounts for any baseline or endogenous $A\beta$ levels in the tissue or medium. The only difference between the slices is the treatment condition, allowing us to determine differences as a result of exogenous $A\beta$ application only. Further, none of the cases used in this experiment had $A\beta$ plaques identified on pathology screen.

12) R2: "In addition it is not enough to cite the literature for the oligomeric Ab composition of the applied brain homogenate, because each human brain sample might differ in terms of Ab species. Therefore, each donor for brain homogenates needs to be analyzed and the species composition determined."

Thank you for raising this important point. We have now performed a characterisation of the different $A\beta$ species present in the homogenate using ELISA and immunogold electron microscopy. This data has been added to the results text (**lines 296-301**) and is summarised in **Supp. Fig. 6**.

13) R2: "One major concern is that the obtained data are over-interpreted and sometimes even misinterpreted. All raw-data displayed in Supplementary Figure 1 and 3 show no significant differences. Only when normalized, differences were significant in some instances. If one looks however closer, it becomes obvious that significance is often only reached due to 1

single case (e.g. Figure 4 c and d vs. raw data supplementary Figure 3).” We apologise if the data analysis and statistical methods were unclear. The statistical modelling used (generalised linear mixed effect models) is a method that allows the factoring in of data distribution to reduce the impact of outliers on the overall statistical model, and therefore the statistical outcome. We have also accounted for other factors within the model, such as case ID, to compare the *relative change* in response to treatment, rather than *absolute change* in the same way that a repeated measures test would do. This ensures that the proportional effect of treatment can be seen even in a data set with large variability in baseline characteristics between cases (e.g. we can detect a 50% increase in A β release regardless of whether that brain sample naturally releases 1pM A β at baseline or 100pM at baseline). The statistics were always done on the raw data throughout the paper unless stated otherwise in the legend (Figure 5 is the exception, explained in detail below). To address this and ensure the assumptions of the statistical test were met, the raw data was transformed into a binary readout. For ease of visualisation, we have shown the control normalised graphs as a percentage for figure 4 and subtraction normalised in figure 5, to allow easier comparisons of the effect on each case for the reader. Therefore, the statistics shown on figure 4 are the same statistics that would have been done on supplemental figure 4. We have added the statistics onto the supplemental datasets to emphasise this point.

For Figure 5, and specifically for Figure 5 b, c & d, due to the heteroskedasticity and zero-inflated nature of the raw data, the data was binarized based on if there was a greater than 2-fold increase in tissue-bound A β compared to medium-only control. This was to ensure that the assumptions of the statistical test were met. Greater than 2-fold increase = increase in A β pathology. < 2 fold increase = no increase in A β . The data was then analysed using a generalised linear mixed effect model with a binomial distribution. Therefore, outliers, whilst impacting the appearance of the graph, do not affect the statistical analysis. For the rest of Figure 5, the statistics performed were done on the medium control subtracted data as the medium control was only assessed as a baseline control for each case and the differences we were interested in were between A β positive and negative conditions. This has been emphasised in Supp. Fig. 7, with dashed lines separating the medium control to the treatment conditions.

14) R2: “The Table with details about the cases on page 14 lists in most of the cases as reason for surgery glioblastoma. Therefore, it should be at least mentioned in the discussion that cases are not from healthy but diseased brains with glioblastoma which are known to display immunreactions that could in turn influence or have an impact on the results.”

Thank you for raising this important point. We have added text discussing limitations of the model (including this point) to the discussion (lines 519-531).

Reviewer 3

15) R3 Summary: “In this manuscript, the authors used live human brain slice cultures (HBSC) from several patients undergoing surgery to remove primary or metastatic brain tumors. During this process, the surgeon resects non-malignant brain tissue to access the tumor. The authors use this access tissue to keep slices in culture and perform experiments to study processes related to Alzheimer’s disease (AD), focusing on the release of amyloid beta peptides, tau, and the relationship of these with known risk factors for AD, namely APOE status and age. The paper could be divided into 1) the basal status of these variables in the culture system, and 2) testing how amenable the HBSC are to pharmacological interventions to study the effects of amyloid beta (Ab) in the brain, particularly on synapses, and

potentially use them as a translational tool. I would like to congratulate the authors for their work. The manuscript is well written, the findings are presented in a logical manner, the results are interesting and overall the quality of the paper is very good (I am particularly impressed by the statistics employed). The many challenges that pre-clinical models of dementia/AD face make this study timely and of critical importance. I think such models could be very informative when testing the mechanisms behind novel therapies. I have a few comments aiming at, in my opinion, increasing the clarity of the manuscript for the reader. They are provided in the same order that the data is presented in the manuscript..”

Thank you for your kind feedback and enthusiasm for our work, and the appreciation of our statistical methods.

- 16) R3: “In the introduction, the authors contextualize their work and the importance of translational tools to study AD such as HBSC by citing work done in human samples (cerebrospinal fluid, interstitial fluid, brain imaging) and in in vitro models (mostly rodent AD transgenic models and induced-pluripotent stem cell derived neurons). Previous work done in human brain slices goes uncited at this point, despite of many years of research and optimisation by other groups. A more balanced introduction might help the reader in contextualizing the present findings. Work cited in the discussion could be cited here”

Thank you for this suggestion, we have now added a paragraph in the introduction to earlier highlight the important previous literature (lines 105-112).

- 17) R3: “In Fig.1, the authors show images of several markers in the cultures at 7 DIV to show the preservation of cellular diversity and function in the HBSCs. They showed the presence of neurons, astrocytes and microglia. I wondered if: Did the authors do stainings of slices at different DIV to show how the presence of these cell types might change, or are maintained, during the culture process? For example, markers of different neuronal types (e.g Satb2 for excitatory neurons, VGlut1 or NMDAR1 for glutamatergic neurons). These data would be interesting to understand how neuronal populations with different levels of vulnerability to AD-pathology remain in the slices, opening up the exciting possibility to study the responses to interventions of specific populations.”

Thank you for this point, we agree that it would be interesting to understand how different neuronal populations are impacted over culture. We have only compared the presence of different markers from acute tissue vs DIV 7 (Supp. Fig. 1), which we have now added to the manuscript (lines 136-140). This shows a typical reduction in markers expected from a resected tissue model, but we do see some differences in preservation of some markers (Tuj1 and GAD2). Other markers such as NR1 show more dramatic decreases. Whilst this highlights limitations of some subpopulations, this would likely be something that should be assessed based on the experimental question in mind.

- 18) R3: “Did the authors monitor cell viability over the course of the experiment with, for example, an LDH assay? Knowing the health of the culture at the time of measurement of the release of tau and other proteins would be important for the interpretation.”

For our work, we had not screened all cases using an LDH assay as this requires a positive control being made from each case, or at least a subset of cases to determine maximal possible toxicity. However, we have since added in an additional 3 cases where LDH assays were performed in addition to having a 1% Triton-x control (to induce maximum culture death

to demonstrate “100% toxicity”), and saw that LDH levels remained low (below 5% cytotoxicity) throughout the culture period (**Supp. Fig. 1**-results **line 139-140**). For all cases we conduct quality control assessments, the details of which we now provide in (**Supp. Fig. 1**). We also assessed TNF- α release in a subset of samples (n=6), and none of them had detectable TNF- α at DIV7 (threshold below 7.8 pg/ml), suggesting a low baseline state of inflammation (data not shown). This seems to line up with similar assessments of human brain slice culture viability (DOI: 10.26508/Isa.201900305, 10.1074/jbc.M111.298471, <https://doi.org/10.1016/j.jneumeth.2018.05.021>, <https://doi.org/10.1016/j.jneumeth.2023.110055>). These studies have all shown, using different assays for slice health, that slice health in culture is typically maintained, and describe similar exclusion criteria for samples to us.

19) R3: “Particularly relevant for glia, did the authors use any marker of activated microglia (such as CLEC7A) over culture? Comparing the levels of such markers at least at 0 DIV (acute slice) with 7 DIV could give some insight into the behavior of cell types during the culture process (particularly, the potential inflammation induced by slicing the tissue). By studying inflammation-related markers in glia cells and cytokines released into the media, one would know whether the culture undergoes acute inflammation that decreases over time, and dictate what would be the best DIV to start conducting experiments without that potentially confounding factor?”

We have since assessed TNF- α presence in the medium of a subset of cases (n=6), from acute, DIV3 and DIV 7 and found that no samples in control conditions had detectable TNF- α release (threshold 7.8 pg/ml) (data not shown). We were able to detect TNF- α release in cases with 1% Triton-x treatment. As glia were not the primary focus of this study, we have not looked at any further markers, but believe that characterisation of slice inflammatory responses over time in culture would be an interesting follow up.

20) R3: “Fig.1g: I found it difficult to see the astrocytic endfeet at this magnification and without a marker of endothelial cells in the vessel. Providing a higher magnification image would be helpful.”

Thank you for this suggestion, we have now updated the image to be of a higher magnification (**Fig. 1g**).

21) R3: “Fig 1h,i: Since electrophysiology is not within my areas of expertise, I found hard to track the data herein presented: Are the synaptic activity and action potentials showed from a representative neuron? How many slices and donors were tested? How many cells? I could not find any of this in the figure legend nor the material and methods. A more integrated data reporting within the manuscript would aid in data interpretation for the reader (and perhaps be helpful for addressing comment 6,d below).”

We have since updated the beginning of the results text (**lines 128, 136,140**) to reflect the study performed. We characterised a subset of cases on both immunohistochemical markers and electrophysiological markers and this has now been added into the results text. We then used the characterisation to confirm our slice cultures were viable, and did not follow this up in further cases to ensure tissue availability for experiments. We have also updated **Table 1** to highlight which cases were used in this experiment.

- 22) R3: “Fig2b,c: There is a remarkable outlier in Ab42 levels (a female case). The authors point in the discussion to a case with very high levels of Ab42, which I think might be this one, having also the higher burden of plaque or amyloid pathology. As the authors discuss, it is indeed intriguing that this case would have the highest levels of soluble Ab42, opposite to what one would expect based on other models/human in vivo samples. Do the authors have any data to investigate if this might be due to biological or technical variability? For example, is there information on the cases’ clinical data that might help in identifying possible reasons (such as epilepsy, cognitive assessments...)? Could they rule out possible technicalities affecting the Ab42 levels of that sample (infection, inflammation...)?”

This case has a significant amount of pathology (both A β plaques and tau tangles), and is one of our older patients in the cohort with an *APOE* 3/4 genotype. There are no other remarkable co-morbidities reported in the patient data. We have updated our summary table (**Table 1**) to reflect this information so readers can review the pathological burden per case. We have since repeated the A β ₁₋₄₂ response using our multiplex assay and found the same result. Oligomeric A β in the culture medium was below detectable threshold by ELISA (similar to the majority of our samples). On measures of slice health and inflammation, TNF- α was below threshold and LDH cytotoxicity was 0.86%. Assessing across other biomarkers within the study, values were well within the standard range, and protein expression within the slice was also within range. All our cultures are carefully screened for contamination so this is not a possibility here. Therefore, we think the high levels of A β 42 in this case is a genuine finding likely due to advanced age, presence of the *APOE* 4 allele and presence of significant pathology as opposed to a technical outlier. There may still be other variables we have not considered, but we also do not have large amounts of tissue to further probe this.

- 23) R3: “Fig.2i: It seems as the females in the *APOE*3/3 group have lower levels of tau released into the media compared to males? As Age was one of the variables (random effects) included in the model, if I understand correctly, sex-dependent differences would have been found?”

We have now added in statistical testing for effects of sex and *APOE* genotype on protein release into the medium. This has been added in to **Fig. 2 and Fig. 3 (described in result lines 186-189, 190-194)**. We have found a significant effect of *APOE* genotype on tau release. In response to one of your later comments (see below) also found a significant increase in NCAM-1 and TDP-43 in samples from males. We have also added in discussion text to expand on these new findings (**lines 526-528**).

- 24) R3: “Fig.3 shows the effects of pharmacologically modifying amyloid beta levels: Did the authors test a solvent-only control alongside the medium-only control? Avoiding this control might skew the results as is common to have effects by organic solvents (in this case, DMSO for BACE1 inhibitor).”

We ran medium only controls for this study as Phosphoramidon was not dissolved in DMSO and less than 1 μ l/ml of treatment was added to the cultures. In order to test whether different treatments resulted in any toxicity that could be influencing the results, we tested a subset of samples on an LDH assay (n=5) and found no difference in LDH release between treatments: Control mean 6.18% range: 1.65 – 18.24%; Phosphoramidon mean: 6.89% range 1.56 – 17.99%; BACE1 inhibitor mean: 5.62% range: 2.06 - 10.62%). We have added this to the results text (**lines 258-259**) and included it in **Supp Fig. 4i**.

25) R3: “The concentration of phosphoramidon is very high - was this required due to penetration issues into the slice? Did the authors try several doses to find the one that would elicit the best response and minimise off-target effects?”

As tissue is under limited supply, we selected a dose from the literature previously shown to effectively prevent degradation of A β in an *in vitro* system (Eckman 2001), which we have now cited in methods **line 714-716**. We tested for toxicity using LDH and found no difference between control and Phosphoramidon treated cultures (see response 23).

26) R3: “Whilst the data clearly shows a decrease in pre-synaptic density when soluble amyloid derived from post-mortem AD brain was applied (Fig.4), supporting the damage and loss of pre-synaptic compartments previously reported in AD, the data to support the opposite (and that sustains the title of the paper) is limited to the upregulation of SYP mRNA. Notably, the authors did not find an upregulation at the protein level (Suppl. Fig.2). The treatment was applied for 7 days (is this correct?), a timeframe during which one might expect to see an effect at the protein level? The discussion of these data in lines 370-379 could be a place to discuss these observations. Adding a point in the discussion about future plans to understand the impact on pre-synaptic compartments I think would be a nice addition.”

In response to this comment and also comments by reviewer 1 (see also response points 4 & 7), we have performed new experiments to better align comparison between the data from Phosphoramidon treatment and data from AD-brain extract treatment studies. Our findings reveal divergent compensatory responses to increased concentration of physiological versus pathological A β . In new **Fig. 4h-i**, we show that synaptophysin (but not PSD95) puncta are reduced by both BACE1 inhibitor and Phosphoramidon treatment conditions after 7 days of treatment, indicating that deviation from baseline A β concentration in either direction induce alterations to presynaptic puncta. The loss of synaptophysin puncta in Phosphoramidon treated cultures may therefore help explain the discrepancy between upregulated synaptophysin mRNA and lack of upregulation at the protein level in western lysates. We also performed additional experiments assessing the impact of pathological A β -containing AD-brain extract on synaptic transcripts (**Fig. 5g**). Interestingly, in contrast to the increase seen with Phosphoramidon treatment, we found no change in synaptic transcripts in response to pathological A β application, potentially indicating a lack of compensatory response under pathological conditions. Taken together, these new data further highlight a potential hormetic role of A β , as well as divergent responses at the synapse level to small (picomolar) increases in physiological versus pathological A β . We provide expanded discussion of these findings in results **lines 253-256, 258-259, 312-314** and discussion **lines 476-480, 485-493, 495**, and have changed the title to better reflect this.

27) R3: “Further to comment 4c, Sebollela et al (PMID: 22235132) reported a decrease in synaptophysin transcripts in adult human cortical slices exposed to sublethal doses of amyloid beta oligomers (A β Os, 500nM). I am curious about this data in relation to the authors finding in their phosphoramidon-treated slices of an increase in synaptophysin (SYP) mRNA (Fig.3g). These findings are discussed in lines 386-388 but not in relation to the phosphoramidon data. Do the authors think the opposite effects are due to a lower amount of amyloid beta reached by this treatment compared to the concentration of synthetic amyloid oligomers used in the other study? The MOAB-2 antibody could recognise soluble oligomers, and they do detect and increase in MOAB-2 positive area, so I would expect and

increase in soluble oligomers with phosphoramidon treatment. Since the authors measured the levels of amyloid peptides in the culture media in these experiments (presented in Supp Fig3 as normalised datapoints), I wondered if they could include the expected final amounts of amyloid peptides reached that elicited the found effects on synaptic transcripts to facilitate comparison to other studies. To specifically find out the final concentration of oligomeric amyloid, they could use ELISA kits with the MOAB-2 antibody as capture and detection if there was an opportunity to do so (<https://uk.vwr.com/store/product/17799539/oligomeric-amyloid-a-elisa-kit>.”

Throughout our study, we work with very low, physiological concentrations of A β . The average concentration released by the control samples in the study in Fig 3 (now Fig 4) was 12.6 (+/0 2.8 SD) pM A β 40 0.46 (+/-0.12 SD) pM A β 42 with the average for Phosphoramidon treated samples being 15.6 (+/- 4.6 SD) pM A β 40 and 0.69 (+/- 0.3 SD) pM A β 42. This is clearly much lower concentration than the 500nM synthetic oligomers used by Sebollela et al. We believe concentration of A β is likely to be highly important, and is likely at least partly responsible for the differences we see between the two studies. We have now further characterised the response of our HBSCs to A β containing AD-brain extract and, despite seeing a reduction in synaptic puncta, we do not see changes to synaptophysin at the mRNA level (data shown in Fig. 5). We have expanded our discussion to provide more direct comparison between our findings and those in Sebollela (see lines 501-506).

As suggested, we attempted to detect oligomeric A β using a MOAB-2 ELISA kit from our samples used in Fig 3 (Control vs Phosphoramidon vs Bace Inhibitor). Oligomeric A β was only detectable (and even then barely above threshold of <0.031 ng/ml) in 9/42 samples with no correlation between treatment type or case where the levels exceeded threshold. We cannot therefore make any conclusions about the specific concentrations of oligomers and how these were impacted by treatment.

- 28) R3: “Fig.4 shows the effects of amyloid beta enriched or depleted extracts from AD brains on synapses using array tomography. They find that whilst amyloid co-localised more often with post-synaptic terminals, the number/area covered by those (PSD95) did not change. Instead, the number of pre-synaptic terminals was reduced. This is an important and interesting finding. Beyond the basics, such as finding it an appropriate imaging technique to study synapses, a detailed critical revision of the array tomography data herein presented lies beyond my expertise. Do the authors have any characterization of the AD-soluble extract preparation to show, such as western blot? Even if supplementary, there is a lack of characterization of the material used, and I think is important to discern the impacts of soluble (monomeric, oligomeric...) and insoluble amyloid on synapses.”

Thank you for highlighting this. We now provide characterisation of the different A β species present in the AD-soluble extract using ELISA and immunogold electron microscopy. We were unable to perform Western blot for characterisation as the concentrations of A β in our sample are exceptionally low (pM level). A β is difficult to detect cleanly on western blot even when loaded at high concentrations and our limited supply of extract prevented us from concentrating the sample to achieve this. Our ELISA and electron microscopy findings are now reported in results lines 296-301 and summarised in Supp. Fig. 6.

- 29) R3 “It's intriguing that despite a decrease in pre-synaptic terminal area, there was no change in the synaptophysin levels in the lysate. Did the authors blot for other pre-synaptic proteins? Perhaps western blot of the whole homogenate might not be sensitive enough. Alternatively, the authors suggest this might be due to a re-distribution of the pool of

synaptophysin protein rather than affecting its levels. If there was material available, the authors could do a fractionation experiment to isolate a synaptosomal-enriched fraction and check whether the levels of such fraction are decreased in Ab+ compared to Ab-, and if the levels of synaptic proteins are disproportionately affected.”

We agree that Western blot analysis is not as sensitive as Array Tomography when assessing levels of synaptic proteins and will also fail to detect reorganisation of proteins away from the synapse. We have also now blotted for Synapsin 1 and SAP97 as further pre- and post- synaptic proteins, and have included this data in **Supp. Fig. 8**. Interestingly, we see the same effect on SAP97 as we did on PSD95, but with no alteration in Synapsin 1 levels. We think that perhaps there is some compensatory response in postsynaptic proteins to the presence of AD brain extract (regardless of A β status), that may result in greater levels of postsynaptic protein (without impacting the number of synaptic puncta). Further research into this phenomenon would be of interest in future studies. Unfortunately, we however do not have enough sample material to do a synaptosomal preparation from the samples – all available tissue had been used for Array Tomography, whole-lysate Western blot or qPCR.

30) R3: “In Fig.5 and Fig.6, the authors show data regarding the endogenous and spontaneous presence of pathological changes linked to AD in the brain slices, namely the aggregation of amyloid species and the phosphorylation of tau at AD-specific sites. The authors show evidence of their presence in the slices and their maintenance after 7DIV, which is appreciated. As expected, those increased with age. Intriguingly and interestingly, they report an increase in Ab42:40 in those slices with higher plaque burden. This is contrary to what one might expect based on the literature in CSF and even ISF. I agree with the discussion (lines 350-352) about analyzing fluid samples and slices derived from the same patient in order to understand this phenomenon better in future work. A few comments: Since none of the cases had been diagnosed with a neurodegenerative disease, and given the relatively young age of the cases, would it not be difficult to discern between early pre-symptomatic dementia, or the natural appearance of amyloid and tau in aging individuals (such as reported by Elobeid et al, 2016)? Compared to studies in NPH patients, which have a higher incidence of dementia (one in two cases) than the expected in the rest of the population, the cases used in this study are not expected to be enriched in early, pre-symptomatic AD cases.”

This is an important point that we have now expanded on in the discussion (lines 519-531). There is likely to be a mixture of benign pathology in aging individuals and pathology that may be pre-symptomatic in our cases. Unfortunately, due to the poor long-term survival of individuals with brain cancer, long term follow up to determine whether someone will go on to develop dementia is unlikely. Despite this, we believe it is useful to look at how the presence of these pathological features (even without a dementia diagnosis) impacts the production and release of proteins into the culture medium as we have done in this paper.

31) R3: “Fig.5b,c: Did I understand correctly that there was no case in which there was co-occurring fibrillary Ab and phospho-tau detected? Since none of the cases had been diagnosed with AD, one might not expect it, but wanted to make sure. To aid the reader, a table summarizing the cases and their characteristics (age, sex, others) related to the amyloid- and tau-pathology status showed (and any potential overlap) would be helpful.

All cases have now been screened and **Table 1** has been updated to include pathology status to help facilitate these interpretations. Across the whole cohort, 3 cases had evidence of both amyloid and tau pathology.

32) R3: “ Since effects on synapses might be expected early in the pre-clinical phase of AD, I am curious to know if the authors performed any synaptic staining in these slices to see if there is any early effect? I also wondered if they have looked for the presence of dystrophic neurites.”

We assessed a subset of cases which had amyloid plaque pathology co-stained with MAP2. We found no clear evidence of dystrophic neurites in these cases (data not shown). This could be due to it being pre-clinical stages of AD, or perhaps even just spontaneous pathology not related to AD as highlighted previously. We did not perform any synaptic staining in these cases past the Array Tomography staining data and so could not explore whether there were any early effects around plaques seen.

33) R3: “Since the authors studied the electrophysiological characteristics of the slices in Fig.1, I wondered if they could mine that dataset for the slices with and without amyloid/tau pathology and see if there are early signs of decreased synaptic activity?”

We have only performed electrophysiological analysis in our characterisation pilot of this study. We therefore have not been able to mine the dataset for changes in synaptic activity in relation to pathology, although this would be an interesting future avenue of research.

34) R3: “The authors discuss the possibility that their ELISA kits are not sensitive enough to detect decreases in Ab42 with age (line 350). Would an MSD assay be more suited to study these samples? I think it has lower LLOD and might be more sensitive to small changes.”

Unfortunately, we were limited with the amount of medium we had from each of the samples to further test numerous analytes. To respond to this point, and the below point, we were able to run a small amount of medium on a multiplex Luminex assay, which also gave us measures of A β ₁₋₄₂ and tau. We found that AB₁₋₄₂ was detected in a very similar range and correlated to that from our ultrasensitive ELISA kit, and did not provide an improvement on sensitivity (data not shown).

35) R3: “If there was media left from these experiments, I would be curious to understand how HBSC recapitulate other early pre-clinical AD processes. Would be great to study the levels of other fluid biomarkers in the media (e.g, NfL, synaptic health markers such as neurogranin...) and their relationship with DIV, age of the donors and Ab levels.”

Thank you for this excellent suggestion. We were able to run all samples from the patient characteristics cohort (the data displayed in **Fig. 2** n=21) on a multiplex assay (Luminex) which included putative neurodegenerative biomarkers KLK-6, NCAM-1, Neurogranin and TDP43. We have now added this data to the manuscript as **Fig 3 & 7**. We show that KLK-6, NCAM-1 and Neurogranin are significantly modulated by brain region, KLK-6 is modulated by *APOE* genotype and NCAM-1 and TDP-43 also show significant sex effects on concentration in the medium (**Fig. 3**). We also further assessed these markers in relation to pathology and show that tau pathology significantly modulates KLK-6 concentration (**Fig. 7**). We think these findings strengthen the paper to expand on how this model can be used to explore pre-clinical AD processes.

Response to reviewers comments: NCOMMS-24-12208A

Reviewer #1

1) R1: "The authors took substantial effort to address my comments. Several new data sets have been acquired and added to the manuscript (array tomography characterisation of human brain slice response to BACE1 inhibitor/ phosphoramidon treatment, qPCR analysis of transcript changes in response to pathological A β -containing AD brain extract, analysis of additional proteins in culture medium, examination of sex effects, additional analysis of relationships between neuronal/synaptic protein and A β / tau release), which strengthen the paper."

Thank you for your kind summary of our additional work. We are pleased that you are satisfied with our response to review and feel the paper has been strengthened.

2) R1: "In the first version it remained somewhat unclear whether viability of slices/neurons was only assessed during the initial experiments or whether these controls were part of the overall study design and performed throughout the course of the experiments. The authors now clarified this point and added additional quality control data assessing macroscopic appearance of the tissue, LDH release over time as a marker of cytotoxicity and expression of synaptic and neuronal proteins during culturing."

Thank you for your description of our additional viability control data

3) R1: "A β 1-40 was found to be tentatively lower in samples derived from older patients. Were there any obvious age-related differences in synaptic density?" The authors now explored this question by quantifying the levels of synaptic markers synaptophysin and PSD95, and neuronal proteins Tuj1 and PGP9.5 using Western blot in correlation with donor age, excluding an age-dependent alteration of synaptic markers (furthermore, whereas Tuj1 was found decreased, PGP9.5 was not). Furthermore, there was no correlation between synaptic and neuronal protein levels with A β 1-40, A β 1-42 or tau in the culture medium."

Thank you for highlighting our additional analysis

4) R1: "Particularly, given that pathological A β (derived from patients with AD) was reported to result in a loss of presynapses it would be interesting to investigate whether an increase of physiological A β and concomitant enhanced levels of synaptic transcripts translate for example into increased synaptogenesis or protection from synaptic loss induced by pathological A β ." The authors now added new experiments addressing these questions. Interestingly, the set of new data showed a loss of synaptophysin puncta following manipulation of physiological A β levels in either direction. An analysis of effects of pathological A β -containing AD-brain extract on synaptic transcripts, furthermore, did not show any change in synaptic transcripts. These data highlight the complexity of the biological processes. The discussion has been expanded accordingly."

Thank you for commenting on our additional work and its appropriate inclusion in the discussion. We agree it is interesting to see varied responses to changing levels of A β .

5) R1: "I find it hard to judge from the array tomography data if the number of synapses per slice affected by oligomeric A β would be high enough to make a functional impact." Additional electrophysiological experiments could not be added due to limited access to human tissue samples and AD-brain extract. The authors now discuss such experiments as possible directions of follow-up studies."

Thank you for pointing out our additional discussion points.

6) R1: "How did the authors deal with the quite prominent lipofuscin-related autofluorescence in human tissue?"

The respective information has been added to the methods section.

Thank you for commenting on our additional methods information

7) R1: "Could upregulation of transcripts related to presynaptic compartments (following induction of A β release) be part of a repair process?"

The discussion has been expanded to cover these points, particularly in light of the newly added data on modulation of physiological levels of A β and the respective effects on synapses.

Thank you for highlighting our additional discussion points

Reviewer #2

8) R2: "The response to reviewers and the revised manuscript satisfactorily addresses my comments and criticisms raised during the review process. I have no additional concerns."

Thank you for your kind comments. We are pleased you are satisfied with our changes to the manuscript.

Reviewer #3

9) R3: "I would like to first thank the authors for their efforts in addressing the reviewer's comments during the first round of revision. The authors have largely addressed the comments I provided, and I am very pleased that addressing these has strengthened the paper and their conclusions, particularly on the potential hermetic role of Abeta in synapses."

Thank you for your kind words, we are very glad you feel our changes have strengthened the paper.

"I will first summarise the points that were successfully addressed and end with minor points that, in my opinion, would still require some amendments or clarification for the reader."

10) R3: By addressing the concerns raised in response points 16 and 27, I think the new version of the manuscript better contextualises the findings with the relevant previous literature."

Thank you for highlighting our text changes in the introduction, we agree it works better to include these points earlier.

11) R3: "In response to point 18, the authors have now included LDH assay data to support the viability of the cultures. Importantly, they also added LDH data upon treatment with BACE1 inhibitor and phosphoramidon, which I think is an important addition and addresses my comments in points 24 and 25. The authors discuss that due to the limited availability of the tissue, they do not routinely assess viability with this assay to avoid sacrificing valuable slices for the 1% Triton X-100 positive control. I appreciate how the most accurate way to assess maximal toxicity is to have a positive control for each case. However, now that the authors have done LDH assays for several cases, I wonder if they have observed such a big variability between positive controls from different cases? If not, they could consider making a batch from multiple cases and re-use this positive control for all their LDH assays. I would prioritise this type of analysis because it requires little amount of volume and, due to their interest in physiologically secreted proteins, membrane permeability would be an important cofounder to take into account."

Thank you for highlighting our additional LDH controls included in the manuscript. We also appreciate your suggestion for re-using positive controls in future, something that we could definitely consider.

12) R3: "Response points 19, 20, 22, 23, 30, 31, 32, 34: thank you for your efforts in addressing these minor points (including my poorly worded question in point 23)."

Thank you for highlighting our minor changes

13) R3: "Response point 26, also raised by reviewer 1: I agree with the authors that the new data on the modulation of amyloid beta and its effects in synapses in living human brain tissue is of importance and a great starting point to study the role of amyloid in synapses in a translational model. I appreciate that the authors tried measuring the levels of oligomeric amyloid in the samples in point 27, and I appreciate how the low levels in the sample would be a technical challenge. I also think is positive that they added new discussion points to address the differences with previous studies."

Thank you for highlighting our additional study and discussion on modulating A β levels.

14) R3: "Response point 28: Thank you for the characterisation data of the AD-soluble extract. I think this is an important addition to present and future experiments using AD-soluble extract to treat slice cultures to model AD-like processes."

Thank you for highlighting our additional characterisation of the AD-soluble extract. We agree this has strengthened the paper and was a very useful suggestion.

R3: "Minor points I think would need addressing:"

15) R3: "In response to points 17 and 21, the authors have included a small characterisation of how neuronal (focusing on synaptic) proteins change over time in culture (acute vs 7DIV) in two cases of the cohort, adding new data into Supp Fig 1 and manuscript lines 136-140. - The western blots in the red channel at the bottom of Supp. Fig.1c are unlabelled, could the authors please label these?"

Thank you for spotting this- we had the channel labelled in the original figure- but it did not carry over in the PDF conversion. This has now been corrected. In addition, the characterisation is from 12 cases total (not 2 as stated above), but we have, for ease of interpretation, shown representative western blot images from only 2 cases. The full western blots are available in the supplementary data file.

16) R3: “Quantifications show significant decreases in neuronal proteins that are commented in the main text as “modest”. I would disagree with this, particularly for PSD-95, and just conclude that for some proteins, the resection and culture conditions had a significant impact.”

We appreciate your observation regarding the decreases in neuronal proteins, particularly PSD-95. We have removed the word modest from our text in line with this.

17) R3: “Do the authors think that the impact on synaptophysin and PSD-95, seemingly greater for the later, may have had any impact on the results of the paper?”

Thank you for your comment. The repeated measures design we use accounts for the effects of culture conditions on the various outputs we are measuring, ensuring that comparisons are made within the same experimental framework (e.g., all slices at 7 DIV in the BACE1 inhibitor and Phosphoramidon study). However, we agree that the observed loss of some synaptic proteins likely reflects a limitation of the *ex-vivo* model that should be considered when extrapolating findings to *in vivo*. We have added this point to our discussion of limitations of the model (**lines 374-375**).

18) R3: “Could the authors please clarify which cases were used for Fig.1 – they are mentioned as a subset of cases, so I understand must be within the 42 cases in Table 1; however, no case has been allocated to Fig. 1 in this table. Are perhaps the cases attributed to Supp. Fig.1 the ones also used in Fig. 1? (n=11, in line with the numbers mentioned in the main text). In the rebuttal letter these cases are referred to as “characterisation pilot”, which made me think they might be independent of the main cohort?”

Apologies for this oversight, we uploaded an old version of the table which did not have this information included. The correct table is now present in the manuscript, with the cases used in figure 1 now listed.

19) R3: “As a minor point, I have noticed some proteins are named inconsistently throughout the manuscript (e.g, PSD-95 and PSD95; TDP-43 and TDP43).”

Thank you for highlighting this. We have opted to use PSD95 and TDP-43 and have corrected this throughout the paper. We will of course take advice from the editors about which version of protein names they prefer in accordance with house style during proofing.

20) R3: “Response point 29: Interesting that synaptophysin and synapsin-1 show different effects, I wondered what are the authors’ thoughts on this finding? I appreciate experimentally addressing this is beyond the scope of the current manuscript.”

Thank you for your comment. We do not observe differing effects on synaptophysin and synapsin-1 when comparing within the same technique. For example, Western blot analysis of lysates from the slices does not show changes in the protein levels of either synaptophysin or synapsin-1 in response to AD-soluble extract (Supplementary Figure 8). We do see increased in PSD95 and

SAP97 in response to this challenge, which we have speculated on in the manuscript as a potential differential response of pre- and post- synaptic proteins.

21) R3: “Response point 35: I am pleased that the revision of the manuscript provided further insights into how brain slice cultures could be used to study the release of fluid biomarkers of neurodegeneration. I noticed that the authors normalise the levels of the secreted proteins analysed by ELISA or Luminex to the total protein content in the medium and wanted to ask the authors about this. In our research, colleagues and I have also looked for an appropriate way to normalise protein release measured by ELISA to account for the variability in the number of cells between conditions. Measuring protein content in media and lysates in parallel by BCA assay in our hands usually results in high absorbance values in media that do not seem to correlate to total protein differences detected in lysates – so as a proxy for cell confluency, we usually normalise ELISA measures of secreted proteins to the total protein content in the lysates. Others measure genomic DNA in the samples and calculate an approximate number of cells in the culture counting that each cell would have approx 6 pg of DNA (<https://www.qiagen.com/us/resources/faq?id=1bfcd402-f33a-4d01-9090-83f150b47276&lang=en>). The caveat with total protein content in media measured by BCA is that such measurements might be challenging due to the supplements present in media (which might contain proteins at higher concentrations than the secreted proteins by the culture, thereby reducing the sensitivity to the culture-derived proteins), and/or to the colour of the media (as in our case, there is phenol red present). Were the authors concerned by the same issues, and if so, how did they address them? Did they do dilutions of the media to address potential problems with phenol red? A more detailed description in material and methods would aid reproducibility of their techniques.”

We thank the reviewer for highlighting our additional work characterising fluid biomarkers in our samples and for this thoughtful comment regarding normalisation procedures. It is something we consider carefully in our experimental design and is an area we are also always seeking to optimise further. Analysis of protein content of the slice itself, as suggested here, would also be a good way of normalising protein content, but does reduce the number of slices we have available for other assays such as Western blot, immunostaining etc. As we are interested in the levels of specific *secreted* proteins compared to all secreted proteins in the medium, we feel normalising from the medium itself is the most reliable way of achieving this. We do not have major issues with high absorbance from human culture medium (unlike mouse organotypic culture medium where high levels of horse serum present can be overwhelming in this assay). We run our human culture medium samples at a dilution of 1:200- 1:400 and, as such, do not have issues with levels of phenol red impacting the absorbance values either. To increase reproducibility, we have now added a methods section on the exact method for the BCA assay (**lines 607-618**).